# Radial glia integrin avb8 regulates cell autonomous microglial TGFβ1 signaling that is necessary for microglial identity

Gabriel L. McKinsey [1] ✉, Nicolas Santander [2], Xiaoming Zhang[3], Kilian L. Kleemann [4], Lauren Tran[1], Aditya Katewa[1], Kaylynn Conant [1], Matthew Barraza[5], Kian Waddell[6], Carlos O. Lizama [7], Marie La Russa[8], Ji Hyun Koo[1], Hyunji Lee[1], Dibyanti Mukherjee[1], Helena Paidassi [9], E. S. Anton[10], Kamran Atabai [7], Dean Sheppard [7], Oleg Butovsky [4] & Thomas D. Arnold [1] ✉

Microglial diversity arises from the interplay between inherent genetic programs and external environmental signals. However, the mechanisms by which these processes develop and interact within the growing brain are not yet fully understood. Here, we show that radial glia-expressed integrin beta 8 (ITGB8) activates microglia-expressed TGFβ1 to drive microglial development. Domain-restricted deletion of *Itgb8* in these progenitors results in regionally restricted and developmentally arrested microglia that persist into adulthood. In the absence of autocrine TGFβ1 signaling, microglia adopt a similar phenotype, leading to neuromotor symptoms almost identical to *Itgb8* mutant mice. In contrast, microglia lacking the canonical TGFβ signal transducers *Smad2* and *Smad3* have a less polarized dysmature phenotype and correspondingly less severe neuromotor dysfunction. Our study describes the spatio-temporal regulation of TGFβ activation and signaling in the brain necessary to promote microglial development, and provides evidence for the adoption of microglial developmental signaling pathways in brain injury or disease.

Microglia, the resident macrophages of the neural parenchyma, are crucial for brain development and the response to brain infection and injury[1–8]. While progress has been made in understanding the effects of microglia on brain development and function, less is known about the reciprocal process - how the brain microenvironment directs microglial specification and activities. During early embryonic development, microglia invade the nervous system and adopt specific transcriptional signatures associated with their specialized functions at this time[9]. Similarly, in adulthood, microglial gene expression is dynamic and dependent on environmental context. For instance,

[1]University of California San Francisco, Department of Pediatrics and Newborn Brain Research Institute, San Francisco, CA, USA. [2]Instituto de Ciencias de la Salud, Universidad de O´Higgins, Rancagua, Chile. [3]Center for Translational Neurodegeneration and Regenerative Therapy, Tongji Hospital affiliated to Tongji University School of Medicine, Shanghai, China. [4]Ann Romney Center for Neurologic Diseases, Department of Neurology, Brigham and Women's Hospital, Harvard Medical School, Boston, MA, USA. [5]Northwestern University, Department of Neuroscience, Chicago, IL, USA. [6]Sidney Kimmel Medical College at Thomas Jefferson University, Philadelphia, PA, USA. [7]University of California San Francisco, Cardiovascular Research Institute, San Francisco, CA, USA. [8]Stanford University, Department of Bioengineering, Stanford, CA, USA. [9]CIRI Centre International de Recherche en Infectiologie, Univ Lyon Inserm U1111 Université Claude Bernard Lyon 1 CNRS UMR5308 ENS de Lyon, F-69007 Lyon, France. [10]University of North Carolina at Chapel Hill, Chapel Hill, NC, USA. ✉ e-mail: gabriel.mckinsey@ucsf.edu; thomas.arnold@ucsf.edu

during inflammatory injury, microglia down-regulate homeostatic genes and adopt a phagocytic, inflammatory, and reactive state[10–13]. Although transforming growth factor beta (TGFβ) signaling is generally known to play an important role in regulating the acquisition of microglial identity[14], and signaling via the TGFβ receptor TGFBR2 regulates microglial homeostasis in the brain[15,16], precisely how TGFβ and other signaling pathways are integrated to mediate microglial development and homeostasis is poorly understood. Defining the molecular processes that underlie microglial differentiation and context-specific state changes is necessary for a more complete understanding of how microglia influence neural development and inflammatory disease.

A substantial body of evidence has shown that TGFβ signaling is crucial for microglial development and function[14]. Most notably, *Tgfb1* null mice display severe neuropathological changes, including microgliosis, neuronal cell death, and synaptic loss[17,18]. However, these mice also have developmental brain vascular defects, making it difficult to determine whether the microglial defects in *Tgfb1* null mice are due to direct effects of TGFβ1 in microglia, or if they are the secondary result of the vascular dysplasia and hemorrhage seen in *Tgfb1* null mice. TGFβ1 protein is synthesized in a latent inactive complex that requires two key steps to be activated[19,20]. First, latent TGFβ1 is anchored to the surface of one cell type by a so-called milieu molecule, LRRC33 (NRROS). Second, the integrin $\alpha_V\beta_8$ dimer on neighboring cells binds to and activates latent TGFβ1, which can then signal to TGFBR2. Cryo-EM analyzes of $\alpha_V\beta_8$ in complex with TGFβ1 and LRRC33 support a model whereby the release and diffusion of active TGFβ ligand is not necessary for TGFβ signaling; rather, TGFβ1 is positioned by LRRC33 and $\alpha_V\beta_8$ to interact with TGFBR2, reinforcing TGFβ signaling in the cell which expresses TGFβ1[19,20]. In a recent publication[21], mice engineered to express TGFβ1 that cannot be released were found to survive without the lethal tissue inflammation found in TGFβ1 constitutive mutants. This paracrine activation / autocrine signaling model predicts that deletion of *Itgb8* in one cell type primarily effects a neighboring cell type which both presents and responds to TGFβ1. Consistent with this model, *Itgb8, Tgfb1, Tgfb1RGE, Lrrc33* and *Tgfbr2* mouse mutants have largely overlapping phenotypes[15,16,18,22–24]. However, the specific cell types expressing these various signaling components, and the timing of their interactions during brain development remain unclear.

In a previous study, we demonstrated a critical role for integrin $\alpha_V\beta_8$ in presenting active TGFβ to microglia[16]. We found that in the absence of $\alpha_V\beta_8$-mediated TGFβ signaling, microglia are developmentally arrested and hyper-reactive. Furthermore, the presence of these dysmature microglia (and not just the absence of mature microglia) causes astrocyte activation, loss of GABAergic interneurons, and abnormal myelination, aspects of pathology which underlie the development of a severe neuromotor syndrome characterized by seizures and spasticity. We found that this phenotype is entirely due to loss of TGFβ signaling in microglia during brain development, and not vascular dysfunction with brain hemorrhage, because microglia-specific deletion of *Tgfbr2* during early stages of brain development completely recapitulated the *Itgb8* mutant phenotype without evidence of vascular/hemorrhage phenotypes. In this study, we aimed to identify 1) The relevant *Itgb8* expressing cell type(s) that mediate microglial TGFβ activation; 2) The developmental timing of *Itgb8*-mediated TGFβ signaling in microglia; 3) The cellular source and identity of the TGFβ ligand relevant for microglial development and homeostasis; 4) The relationship between developmentally disrupted microglia and disease associated microglia; and 5) The role of canonical (Smad-mediated) versus non-canonical TGFβ signaling in microglia. By systematically deleting *Itgb8* from different cell lineages of the embryonic, neonatal and adult brain, we show that *Itgb8* expression, specifically in embryonic radial progenitors, is necessary for microglial differentiation, but is not required to maintain microglial homeostasis in adulthood. We find that microglia-expressed *Tgfb1* is required cell-

autonomously for differentiation and, in contrast to *Itgb8*, is required to maintain homeostasis. By analyzing various *Itgb8-TGFβ* pathway mutants, we find that non-canonical TGFβ signaling fine-tunes microglial maturation and homeostasis, such that upstream mutants have amplified microglia and neuromotor phenotypes compared to *Smad2/3* conditional mutants. These data support the model that ITGB8/TGFβ1 signaling via radial glia-microglial interactions is crucial for the embryonic maturation of microglia and that in the absence of this signaling, developmentally arrested dysmature microglia drive local neuroinflammation and disruptions in neurological function. Finally, we show that dysmature and disease associated microglia share transcriptional and epigenetic properties, pointing to core TGFβ-regulated gene networks active in both development and disease.

## Results

### Radial glia *Itgb8* expression promotes microglial development

We previously found that central nervous system (CNS)-wide deletion of *Itgb8* using *Nestin^Cre* leads to microglial dysmaturation and associated neuromotor impairments nearly identical to those seen in mice with conditional deletion of *Tgfbr2* in microglia[16]. *Nestin^Cre* constitutively removes *Itgb8* from all cells derived from the early neuroepithelium (astrocytes, oligodendrocytes, and neurons), and *Itgb8* is expressed in all of these cell types (Figure S1A, B)[25]. To determine which of these *Itgb8*-expressing cells are responsible for the phenotypes observed in *Itgb8^fl/fl;Nestin^Cre* mice, we generated six different *Itgb8* conditional knockout lines (*Itgb8^fl/fl;Nestin^Cre, Itgb8^fl/fl;hGfap^Cre, Itgb8^fl/fl;Olig2^Cre, Itgb8^fl/fl;Syn1^Cre, Itgb8^fl/fl;Ald1l1^CreER, Itgb8^fl/fl;Ubc^CreER* to systematically delete *Itgb8* from astrocytes, oligodendrocytes or neurons individually and in combination (Figure S1C). As summarized in Figure S1, none of these cell-type specific *Itgb8* mutants produced the motor dysfunction or cellular phenotypes observed throughout the nervous system in *Itgb8^fl/fl;Nestin^Cre* mice (e.g., reduced TMEM119 and P2RY12 expression; increased CD206, LGALS3 and GFAP expression) (Figure S1C, S2). Most notably, conditional deletion of *Itgb8* using *hGFAP^Cre* resulted in no obvious neuropathological changes despite recombining nearly all neurons, astrocytes and oligodendrocytes in the cerebral cortex (98.3+/−0.3%, 100% and 98.1+/−2.7%, and respectively; Figure S1C, D). These data indicate that *Itgb8* is dispensable for maintaining microglial, neuro/glial and vascular homeostasis in adult mice.

To understand why *Nestin^Cre* mediated *Itgb8* deletion resulted in significant neuromotor dysfunction, while deletion from all other brain cell types (alone or in combination) did not, we considered the unique features of this line[25]. *Nestin^Cre*-mediated recombination occurs at approximately embryonic day 10.5 (E10.5), resulting in loss of *Itgb8* in radial glia and their progeny (Fig. 1C, D and Figure S1). *hGFAP^Cre* also recombines radial glia progenitors and their progeny, but three days later than *Nestin^Cre* at ~E13.5[26,27]. Both of these lines have recombination in almost all cortical neurons and glia by adulthood (Figure S1). The sequential recombination of brain progenitors in *Itgb8^fl/fl;Nestin^Cre* (complete phenotype) and *Itgb8^fl/fl;hGFAP^Cre* (no apparent phenotype) suggests that the developmental timing of *Itgb8* expression in, or deletion from, early embryonic neuroepithelial progenitor cells or radial glia is responsible for the stark phenotypic differences in these mice.

Published bulk and single cell RNA-Seq (Fig. 1A and S3A)[28,29], embryonic Flash-Tag scRNA-seq (Figure S3B)[30], and in situ hybridization (Figure S3C) show that *Itgb8* is expressed in neuroepithelial progenitor cells throughout the brain starting by E8.5, and that by E14.5 *Itgb8* is most highly expressed in radial glia progenitors, and less so in maturing astrocytes, oligodendrocytes and neurons[29,30]. We confirmed this using *Itgb8-IRES-tdTomato (Itgb8^tdT)* reporter mice, finding intense tdT expression in ventricular zone progenitors at E14.5 (Fig. 1B and S3D)[31] as well as tdT expression in radial glia fibers and endfeet in the embryonic meninges (Figure S3E).

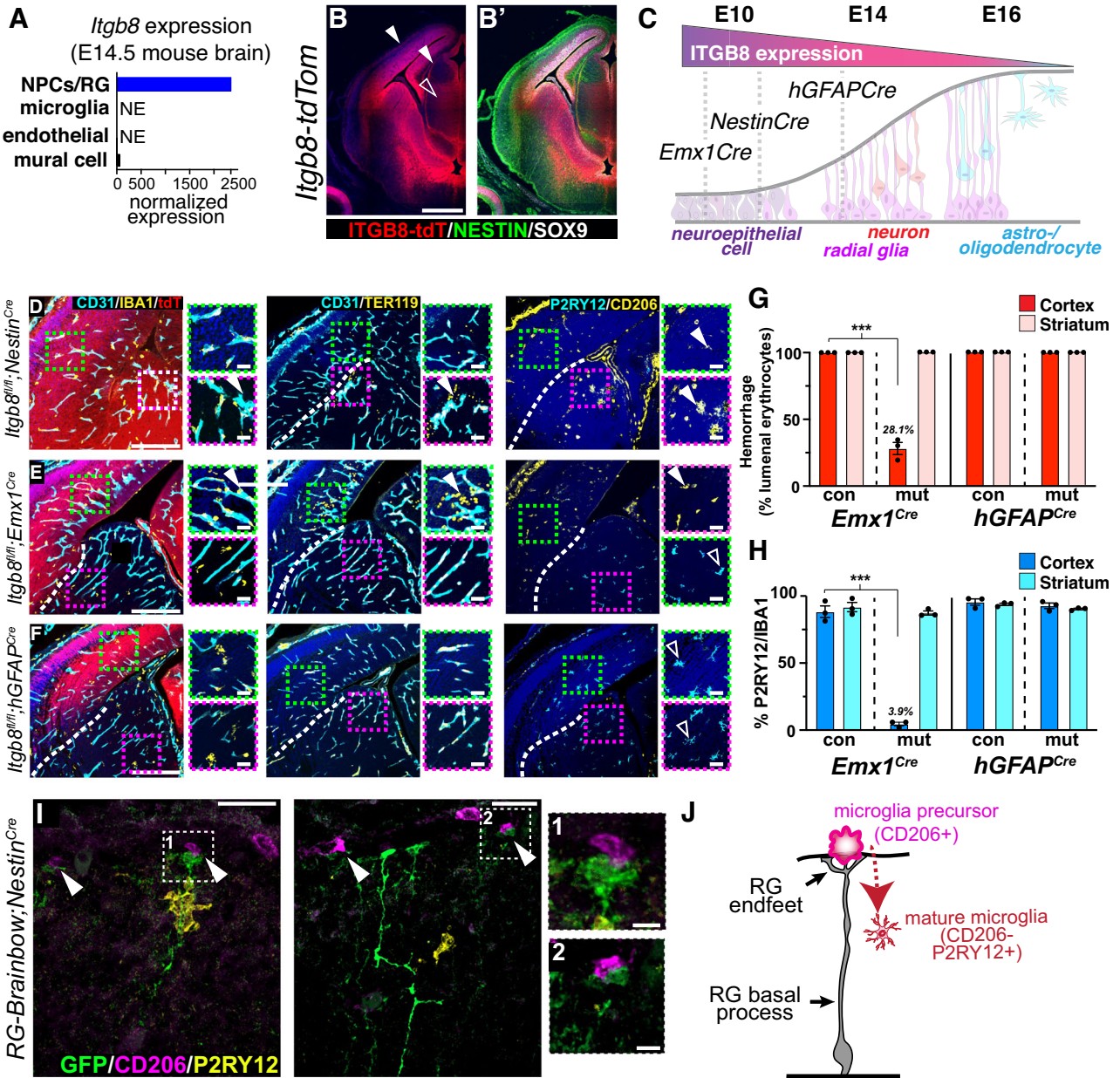

**Fig. 1 | Deletion of *Itgb8* in early embryonic radial glia disrupts microglial maturation. A** Analysis of *Itgb8* expression in the E14.5 mouse embryo in neural progenitor cells (NPCs) and radial glia, microglia, endothelial cells, and mural cells[28]. **B** *Itgb8*[tdT] reporter expression confirms strong *Itgb8* expression in SOX9+, NESTIN+ radial progenitors at E14.5. Open arrowhead marks radial glia fibers; closed arrowhead marks ramified radial glia endfeet at the surface of the neuroepithelium. **C** Model describing developmental expression of *Itgb8* in neuroepithelium and radial glia, and correlation with sequential timing of Cre recombination in *Emx1*[Cre], *Nestin*[Cre] and *hGFAP*[Cre] lines. **D–F** Deletion of *Itgb8* from neuroepithelial and radial progenitors using indicated Cre lines. Coronal brain sections stained for tdT (Cre recombination, red), vascular endothelium (CD31, cyan), and macrophages/microglia (IBA1, yellow); hemorrhage (red blood cells marked by TER119, yellow) observed outside of vascular lumen (CD31, cyan); microglia precursors (CD206,

yellow) and committed/homeostatic microglia (P2RY12, cyan). **G** Quantification of brain hemorrhage in *Itgb8*[fl/fl];*Emx1*[Cre] and *Itgb8*[fl/fl];*hGFAP*[Cre] E14.5 cerebral cortex and underlying striatum. $p = 9.3 \times 10^{-5}$. **H** Quantification of microglial P2RY12 expression in embryonic IBA1+ microglia in *Itgb8*[fl/fl];*Emx1*[Cre] and *Itgb8*[fl/fl];*hGFAP*[Cre] E14.5 cerebral cortex and underlying striatum. $p = 5.1 \times 10^{-5}$. **I** E14.5 brain section from *Emx1*[Cre];*RG-brainbow* mouse stained for membranous GFP (individual recombined radial glia; endfeet, green), microglia precursors (CD206, magenta), and committed/homeostatic microglia (P2RY12, yellow). Arrowheads indicate foot process of radial glia contacting pial-associated CD206+ presumptive microglia precursor (model in **J** to right). $n = 3$ for **B**, **D–I**. Error bars = mean +/- SEM. Scale bar in B = 500μm, **D–F** = 200μm and inset scale bar=30μm, **I** = 25 μm. ***$p < 0.001$, two-tailed t-test. Source data are provided as a Source Data file.

To more directly test the hypothesis that early developmental expression of *Itgb8* is necessary for microglial maturation, we analyzed *Itgb8* conditional knockouts (*Nestin*[Cre], *hGFAP*[Cre], *Olig2*[Cre], *Syn1*[Cre]) at E14.5. We also generated *Itgb8*[fl/fl];*Emx1*[Cre] mice, in which Cre recombinase is expressed in neocortical neuroepithelial cells just before the formation of radial glia at E9[32] (Fig. 1C). As we previously reported[16], we found that *Itgb8*[fl/fl];*Nestin*[Cre] mice develop vascular dysplasia and

hemorrhage in the brain starting at E11.5, which progresses in a ventral-dorsal fashion to include all periventricular vessels by E14.5 (Fig. 1D). Coinciding with these vascular changes, we observed that IBA1+ macrophages were strongly associated with dysplastic vessels in *Itgb8*[fl/fl];*Nestin*[Cre] mutants (IBA1/CD31 apposition in Fig. 1D), and were characteristically dysmature, i.e., lacking expression of mature/homeostatic microglia marker P2RY12, and reciprocally upregulating

or persistently expressing CD206 (MRC1), a marker of undifferentiated microglial precursors and border associated macrophages (BAMs) (arrowheads in right column of Fig. 1D)[33–36]. While the vascular phenotype and hemorrhage was largely localized to areas near the cerebral ventricles (the periventricular vascular plexus, PVP), we observed microglia lacking P2RY12 staining throughout the brain, including areas far from hemorrhage, suggesting that the two phenotypes might be dissociable, i.e., that hemorrhage is not causing these widespread microglial changes or vice-versa.

Similar to *Itgb8fl/fl;NestinCre* embryos, *Itgb8fl/fl;Emx1Cre* E14.5 embryos had vascular abnormalities and hemorrhage near the cerebral ventricles, concomitant with microglial changes throughout the developing cortex and hippocampus (Fig. 1E, controls in Figure S4A-F). However, these vascular/hemorrhage and microglial abnormalities in *Itgb8fl/fl;Emx1Cre* mice were strikingly domain-specific; hemorrhage (as the percentage of RBCs inside of blood vessels, Fig. 1G) and microglial dysmaturity (%IBA1+ cells lacking P2RY12, Fig. 1H) occurred only in the developing cerebral cortex and hippocampus, where Cre recombinase had been active (as indicated by tdTomato (tdT) Cre-reporter expression), but not in the underlying striatum (purple boxes in Fig. 1E, quantification in Fig. 1G, H). In sharp contrast, *Itgb8fl/fl;hGFAPCre* mice lacked any apparent vascular or microglial phenotype (Fig. 1F–H). Similar to *Itgb8fl/fl;hGFAPCre* mice, we observed no obvious vascular or microglial phenotype in *Itgb8fl/fl;Olig2Cre* or *Itgb8fl/fl;Syn1Cre* mutants (Figure S4G–K). These Cre lines are active at or before E14.5, but not in radial glia. To validate the deletion of *Itgb8* using the *Emx1Cre* and *hGFAPCre* mouse lines, we performed western blot analysis of *Itgb8fl/fl;Emx1Cre* and *Itgb8fl/fl;hGFAPCre* embryonic cerebral cortex (Figure S4L, M). This analysis showed significant reductions in ITGB8 protein expression in the E14.5 *Itgb8fl/fl;Emx1Cre* (85.5+/−3.3% reduction) and *Itgb8fl/fl;hGFAPCre* (58.2+/−14.1% reduction) embryonic cortex. The incomplete and different degrees of protein reduction in these mutants is likely related to timing of Cre expression relative to radial glia differentiation: *Emx1Cre* recombines earlier and in more downstream progeny than *hGFAPCre*[27]. Based on the pattern and timing of both *Itgb8* expression and Cre-induced recombination in these various lines (*Emx1Cre* ~ E9; *NestinCre* ~ E10.5; *hGFAPCre* ~ E13.5, see Fig. 1C and adult recombination analysis in Figure S1), our data indicate that *Itgb8* expression in early stage neural progenitors promotes both vascular and microglia maturation, and that the expression of *Itgb8* in astrocytes, oligodendrocytes, and in post-mitotic neurons is dispensable for brain vascular and microglia development and ongoing homeostasis.

We sought to further examine the interactions between radial glia and microglia. Initial descriptions of microglia by del Rio-Hortega proposed that microglia are developmentally derived from the inner layer of the meninges, the pia mater[37]. While CD206 is a marker of borderzone macrophages in adulthood, recent genetic fate-mapping experiments indicate that microglia are derived from CD206-expressing undifferentiated macrophage precursors[34–36]. Upon differentiating, these progenitors down-regulate the expression of CD206 and up-regulate microglial markers such as P2RY12. To determine whether embryonic radial glia make contact with CD206+ microglial precursors, we crossed *Emx1Cre* mice to a mouse line (*RG-Brainbow*) that expresses a Cre-dependent plasma membrane-tagged fluorescent reporter in a mosaic fashion under the radial glia expressed GLAST promotor[38]. We then examined the interaction between radial glial endfeet and CD206+ macrophages in the meninges, where most embryonic CNS-associated CD206+ macrophages reside. In E14.5 *Emx1Cre;RG-Brainbow* embryos, we observed contact between CD206+ meningeal macrophages and radial glia endfeet marked by membranous mGFP (arrowheads in Fig. 1I). Taken together with our observations above, we hypothesize that meningeal CD206+ microglial precursors interact with *Itgb8* expressing radial glial endfeet and that this interaction promotes TGFβ-dependent differentiation in these progenitors. In the absence of *Itgb8*, CD206+ microglial precursors

populate the brain, but fail to express more mature microglial-specific markers (Fig. 1J).

## Domain-specific microglial dysmaturity persists in adult *Itgb8fl/fl;Emx1Cre* mice

We next followed *Itgb8fl/fl;Emx1Cre* mice into adulthood to see how developmental phenotypes might evolve over time. Similar to *Itgb8fl/fl;NestinCre* mice, vascular defects and hemorrhage resolved, while microglial changes persisted. To explore the hypothesis that dysmature microglia in *Itgb8fl/fl;Emx1Cre* mice are developmentally arrested, we compared their transcriptional phenotypes[39] to those of microglia collected at various developmental stages from an early yolk sac progenitors into adulthood[40]. We found a striking enrichment of dysmaturity genes (those that are differentially expressed in *Itgb8fl/fl;Emx1Cre* versus control mice) in clusters 3-5, which represent transcripts found in yolk sac-derived precursors and early embryonic microglia (Fig. 2A, B). Conversely, genes that were enriched in clusters 6 and 7, which represent late embryonic/neonatal and postnatal stages were strongly downregulated (Fig. 2A, B). Immunohistochemical analysis confirmed that the dysmature microglial phenotype was characterized by loss of homeostatic microglial markers (P2RY12, TMEM119) and upregulation or persistent expression of reactive/immature markers (CD206, CLEC7A and APOE) and was tightly restricted to the *Emx1Cre* recombination domain (Fig. 2C–G, S4N, O). Quantification of P2RY12 expression in the cortex versus striatum of *Itgb8fl/fl;Emx1Cre* adult mice showed a near complete loss of cortical P2RY12 expression and a substantial increase in APOE expression in IBA1+ cortical microglia (Fig. 2G). Outside of the domain of recombination, microglia were comparable to control mice. Surprisingly, and in contrast to *Itgb8fl/fl;NestinCre* mice, *Itgb8fl/fl;Emx1Cre mice* had no obvious neuromotor deficits or early mortality, suggesting that the profound motor disturbances in *Itgb8fl/fl;NestinCre* mice are not solely derived from microglial changes in the motor or sensory cortex.

We found that many of the genes enriched in adult dysmature *Itgb8fl/fl;Emx1Cre* microglia are similarly enriched in various disease and injury states (DAM, MGnD)[11,17] (Fig. 2A, H). Likewise, many disease and injury-associated genes up-regulated in adult *Itgb8fl/fl;Emx1Cre* microglia are developmentally expressed (Fig. 2A), supporting the model that disease and injury-associated microglia recapitulate transcriptional states associated with progenitor and early embryonic developmental stages[41,42]. In addition to these disease and injury associated genes, adult dysmature *Itgb8fl/fl;Emx1Cre* microglia expressed BAM markers (Fig. 2A, H), which are also expressed in developing microglia. Together, these experiments indicate that domain-specific deletion of *Itgb8* in radial glia progenitors leads to a zonally restricted arrest in microglial development. These developmentally arrested microglia appear to remain trapped in both time (an early embryonic progenitor state) and space (bounded by progenitor Cre recombination domains), and share transcriptional properties with disease and injury-associated microglia and borderzone macrophages.

## Dysmature microglia show pervasive epigenetic changes associated with MGnD and BAM gene upregulation

To better define the mechanistic basis for the transcriptional changes in *Itgb8* mutant dysmature microglia, we performed assays for transposase-accessible chromatin with sequencing (ATAC-seq) to identify regions of differentially accessible chromatin in *Itgb8fl/fl;Emx1Cre* dysmature microglia versus microglia from control brains (Fig. 3). We also performed ChIP-Seq assays of histone H3 lysine 9 acetylation (H3K9ac) to identify the changes in active promoters and enhancers in *Itgb8fl/fl;Emx1Cre* dysmature microglia. We found a moderate linear correlation between genes associated with ATAC-seq defined differentially accessible peaks (DAPs) and *Itgb8fl/fl;Emx1Cre* dysmature microglia gene expression (R = 0.42, p < 2.2e−16) (Fig. 3B) and strong correlation between genes associated with differential

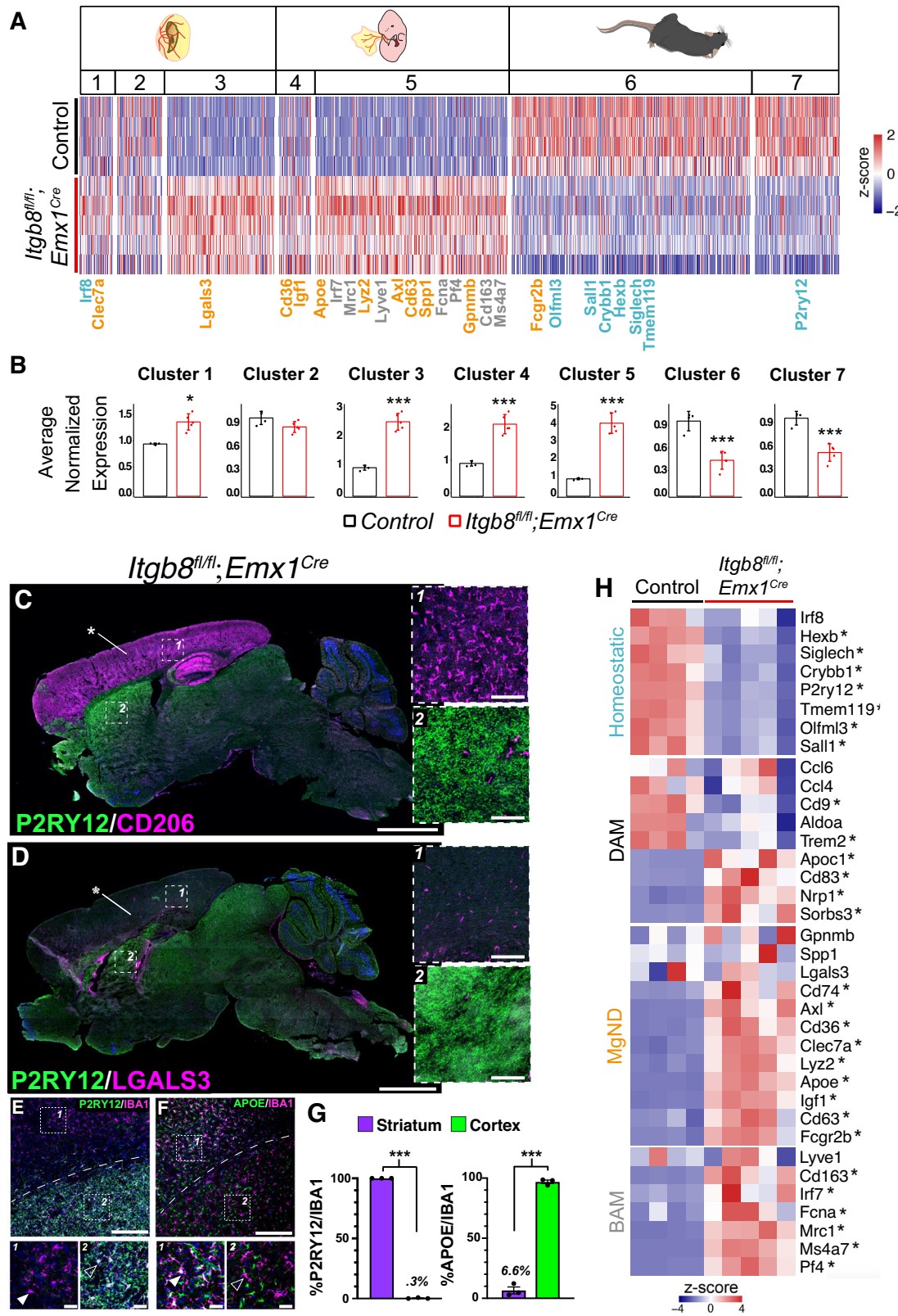

enrichment of H3K9ac ChIP-seq peaks (DEPs) in dysmature microglia and changes in dysmature microglia gene expression (R = 0.82, p < 2.2e$^{-16}$) (Fig. 3C). Of the genes with the most significantly reduced levels of open chromatin and H3K9ac enrichment, many were markers of microglial identity (*P2ry12*, *Hexb*, *Sall1*) (Fig. 3 and S5A, D). In contrast, genes with the most significantly increased levels of open chromatin and H3K9ac enrichment were largely BAM (*Mrc1*, *Pf4*, *Ms4a7*) or

MGnD markers (*ApoE*, *Clec7a*, *Axl*) (Fig. 3, S5B-D). A comparison to a previously published ATAC seq analysis of developing microglia[43] revealed that *Itgb8*$^{fl/fl}$;*Emx1*$^{Cre}$ dysmature microglia share epigenetic profiles most similar to yolk sac progenitors and early embryonic immature microglia (Figure S5E). Many of these microglia identity genes are directly regulated by SMAD4, and many BAM or MGnD genes are ectopically bound by SMAD4 in mice lacking *Sall1* expression in

**Fig. 2 | Domain-specific microglial dysmaturity persists in *Itgb8^fl/fl^;Emx1^Cre^* mice.**
**A** Comparison of the transcriptional properties of adult *Itgb8^fl/fl^; Emx1^Cre^* mutant and control *Itgb8^fl/fl^* microglia using stage specific developmental markers. *Itgb8^fl/fl^; Emx1^Cre^* mutant dysmature microglia retain the gene expression profiles of early embryonic microglia, and lack expression of mature homeostatic microglial genes (teal)[40]. Heatmap shows z-score of normalized expression. Many of the early embryonic marker genes enriched in dysmature microglia in *Itgb8^fl/fl^; Emx1^Cre^* mutants are also MgND (orange) and BAM (gray) markers. **B** Analysis of developmental gene cluster expression in *Itgb8^fl/fl^; Emx1^Cre^* mutant microglia reveals enrichment for progenitor (cluster 1) and early embryonic phase (clusters 3-5) enriched gene sets. Graphs indicate mean ± SD of the average normalized expression in each cluster. $*p < 0.05$, $***p < 0.001$; two-tailed t-test. **C** Whole brain sagittal immunostaining of adult *Itgb8^fl/fl^; Emx1^Cre^* mice revealed anatomically restricted maintenance of the microglial precursor marker CD206 in the cortex and hippocampus (asterisk and box 1), vs striatum (box 2). **D** Increased MGnD marker LGALS3 expression in cortical and hippocampal microglia in *Itgb8^fl/fl^; Emx1^Cre^* mice (asterisk and blow-up in box 1), and no change in the underlying striatum (blow-up in box 2).

**E** Downregulation of the homeostatic marker P2RY12 in the cortex of a *Itgb8^fl/fl^; Emx1^Cre^* mouse, and maintenance in the underlying striatum. **F** Cortex-restricted upregulation of the reactive marker APOE in IBA1⁺ cells of the cerebral cortex, and lack of APOE expression in microglia in the striatum. Closed and open arrowheads in (**E**, **F**) mark dysmature cortical and phenotypically normal striatal microglia respectively. **G** Quantification of homeostatic (P2RY12) ($***p = 2.9*10^{-10}$) and reactive (APOE) ($***p = 9.68*10^{-5}$) markers in the cerebral cortex and striatum of the adult *Itgb8^fl/fl^; Emx1^Cre^* brain. **H** Expression of microglial status markers in isolated microglia. Heatmap shows z-score of normalized expression. Genes with an asterisk are differentially expressed ($*p < 0.05$; DESeq2, negative binomial generalized linear model and Wald test for significance testing). Error bars= mean +/- SEM. $n = 4$ for control and $n = 5$ for *Itgb8^fl/fl^; Emx1^Cre^* mutants in (**A** and **B**). Cx= cerebral cortex; Cc= corpus callosum; Str= striatum; Dashed line= cortical/striatal boundary. Scale bar in **C**, **D** = 2 mm and inset scale bar=200$\mu$m, **E**, **F** =150$\mu$m, and inset scale bar=20$\mu$m. $n = 3$ for 2C-G. $***p < 0.001$, two-tailed t-test. Source data are provided as a Source Data file.

microglia, consistent with direct SMAD4-mediated gene activation or repression initiated by ITGB8-TGFβ signaling[44]. De novo motif enrichment analysis of DAPs most highly increased or decreased in ATAC-seq or H3K9ac-seq recovered motifs recognized by PU.1, MEF2C, RUNX1, SMAD2, and MAFB, all of which are transcription factors known to be relevant for microglial development and homeostasis[14,45–47] (Fig. 3F). The changes in the availability of these gene regulatory elements in *Itgb8^fl/fl^;Emx1^Cre^* cortical and hippocampal microglia are therefore likely to reflect widespread changes in the binding and regulatory activity of transcription factors that are crucial for the acquisition and maintenance of microglial identity.

## *Tgfb1* is required cell-autonomously for microglial development

The only known biological function of ITGB8 is to activate TGFβ1 and TGFβ3[19,20]. In line with this, *Tgfb1^−/−^* mutant mice display neuropathology that closely resembles the various neurovascular and neuroimmune phenotypes observed in *Nestin^Cre^;Itgb8^fl/fl^* and *Emx1^Cre^; Itgb8^fl/fl^* conditional mutants[18]. Recent structural biology studies[19,20] reveal a mechanism of TGFβ1 activation where mature TGFβ1 signals within the confines of latent-TGFβ1; the release and diffusion of TGFβ1 is apparently not required. This paracrine activation / autocrine signaling model predicts that deletion of *Itgb8* on one cell type primarily affects another, and that deletion of *Tgfb1* will only affect the cells from which it is deleted. Here, we looked to test this model as it applies to microglial development.

To determine the developmental expression of *Tgfb1* in the brain, we examined a published RNA-seq dataset, which showed strong expression of *Tgfb1* in brain macrophages, and weaker expression in brain blood vessels (both endothelial cells and PDGFRβ⁺ mural cells) at E14.5 (Fig. 4A)[48]. Within the myeloid compartment, published bulk RNA-seq from sorted microglia and BAMs[33] indicates that *Tgfb1* is highly expressed in both microglia and BAMs throughout development (Fig. 4B). In contrast, *P2ry12* and *Pf4* are specific markers for these cell types, respectively, and are expressed at early developmental stages. To test the hypothesis that cell-autonomous TGFβ1 is necessary for microglial development, we generated *Tgfb1^fl/fl^;Cx3cr1^CreER^* (Fig. 4C, D) and flox/null *Tgfb1^fl/GFP^;Cx3cr1^CreER^* E14.5 embryos from dams treated with tamoxifen daily for the three days prior to collection (E11.5-E13.5)[49,50]. RNAScope in situ hybridization showed strong expression of *Tgfb1* in IB4 (Isolectin B4)⁺ blood vessels, IB4⁺ microglia and BAMs (meningeal and choroid plexus macrophages), and loss of *Tgfb1* expression in microglia and macrophages in *Tgfb1^fl/fl^;Cx3cr1^CreER^* E14.5 embryos (Fig. 4C, D).

To specifically delete *Tgfb1* in BAMs or microglia, we relied on our previous findings that *Pf4^Cre^* and *P2ry12^CreER^* specifically drive recombination in these populations respectively[36]. After finding that *P2ry12^CreER^* was unable to efficiently delete *Tgfb1* in all microglia, we

sought to improve deletion efficiency using homozygous *P2ry12^CreER^* mice. We generated *Tgfb1^fl/GFP^;P2ry12^CreER/CreER^;Ai14* E14.5 embryos, inducing three days prior as with *Cx3cr1^CreER^* mice. Importantly, *P2ry12^CreER^* is a P2A fusion knock-in allele that retains P2RY12 protein expression in homo and heterozygous mice[36]. In contrast to *P2ry12-CreER* heterozygotes, which showed a partially penetrant phenotype, *Tgfb1^fl/GFP^;P2ry12^CreER/CreER^;Ai14* (Fig. 4E, F) E14.5 embryos had a near complete loss of P2RY12 expression in IBA1⁺ microglia (0.6+/−0.6% versus 90.8+/−1.0% in *Tgfb1^fl/fl^* controls) that was as pronounced as *Cx3cr1^CreER^*-mediated *Tgfb1* deletion (5.0+/−2.2%) (Fig. 4J), although meningeal macrophages were not recombined (open arrowheads in Fig. 4F, H). These data suggest that the microglial phenotype observed in these models is due to loss of *Tgfb1* in microglia, and not microglia progenitors or BAMs. To more directly test the function of *Tgfb1* in BAMs, we generated *Tgfb1^fl/GFP^;Pf4^Cre^;Ai14* E14.5 embryos. We previously showed that *Pf4^Cre^* specifically recombines all BAMs including choroid plexus, meningeal and perivascular macrophages[36]. *Pf4* is also highly expressed in yolk sac myeloid precursors[43]. Retention of recombined tdT⁺ cells in border-associated spaces, and lack of recombination in microglia, indicate that *Pf4* is an early marker of BAM-committed cells. Consistent with this, we observed specific recombination of meningeal and choroid plexus macrophages in E14.5 control and *Tgfb1* conditional knockouts, but no significant changes in microglial P2RY12 expression (Fig. 4G, H). Furthermore, none of these microglial conditional mutants showed an increase in brain hemorrhage (Fig. 4I). Together, these data indicate that microglia require self-produced TGFβ1 for their differentiation.

We next looked to determine the source of *Tgfb1* for brain vascular development, and to probe potential interactions between developing brain blood vessels and microglia. We generated endothelial cell (*Cdh5Cre^ERT2^* and *Tie2^Cre^*)[51,52] and vascular mural cell (*Pdgfrb^Cre^*)[53] *Tgfb1* conditional mutants, and compared these to *Tgfb1^−/−^* mutants as a positive control (Fig. 5A, B). As previously documented[54], *Tgfb1^−/−^* mice showed prominent vascular changes (glomeruloid malformations, dilated and tortuous vessels) and associated hemorrhage throughout the cerebrum (asterisk in Fig. 5B). Concomitant with vasculopathy and hemorrhage, microglia in *Tgfb1^−/−^* embryos completely lacked P2RY12 staining, even in microglia distant from hemorrhaging (Fig. 5B). In comparison, *Tgfb1^fl/fl^;Cdh5^Cre^* endothelial mutants (Fig. 5C) and *Tgfb1^fl/fl^;Pdgfrb^Cre^* mural cell mutants (Fig. 5D, H) both retained P2RY12 expression in microglia, while *Tgfb1^fl/fl^;Tie2^Cre^* mutants showed an almost complete loss of P2RY12 expression in these cells (16.0+/−6.9% versus 90.7+/−1.0% in *Tgfb1^fl/fl^* controls) (Fig. 5E, G). We suspect that reduced P2RY12 expression in *Tie2^Cre^* mutants is due to early developmental recombination in, and *Tgfb1* deletion from,

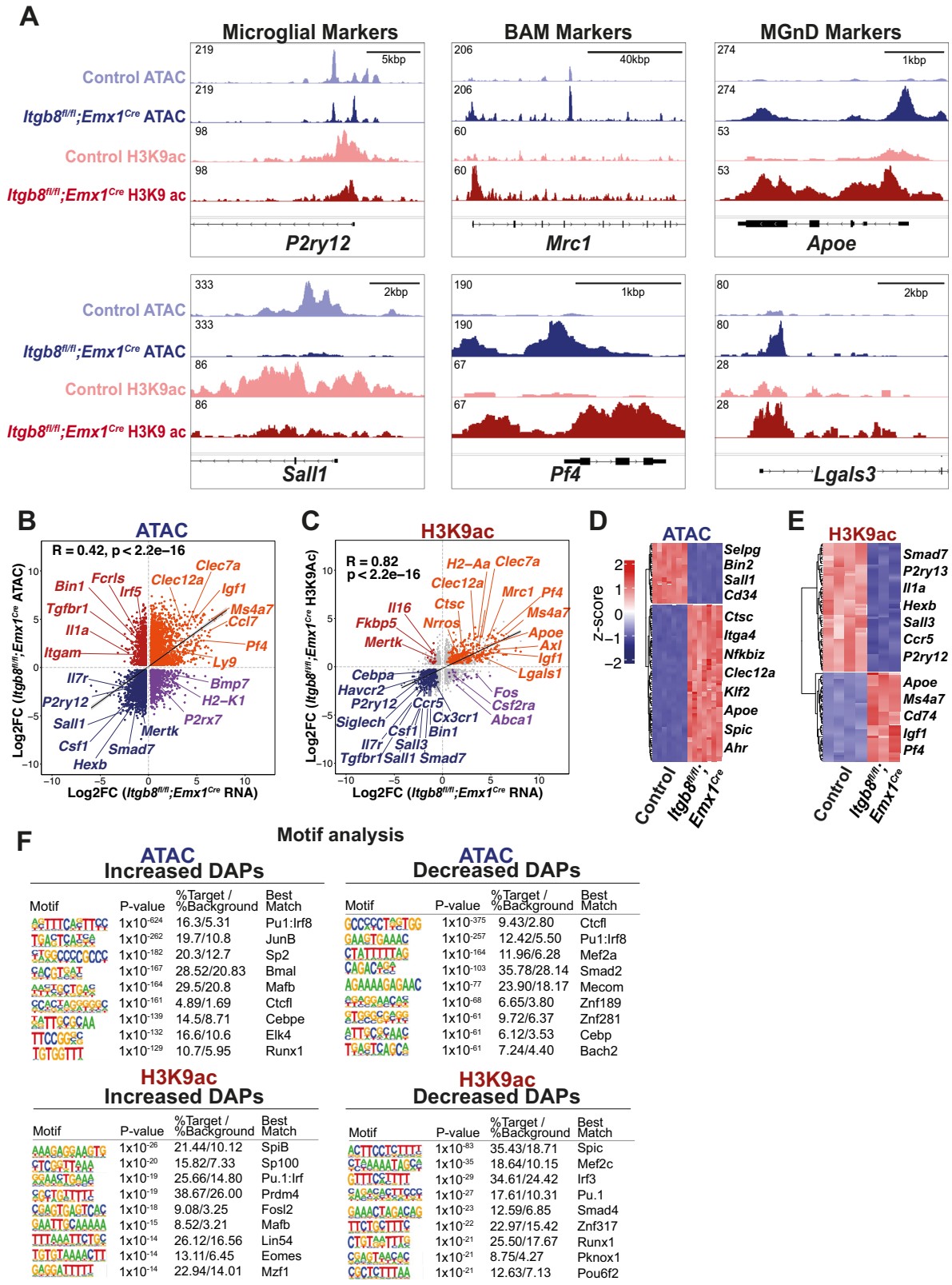

microglia precursors; *Tie2^Cre* is well-known to recombine yolk sac hemogenic endothelium from which microglia are derived.

Surprisingly, *Tie2^Cre*, *Cdh5^CreER* and *Pdgfrb^Cre* mutants had few vascular changes (Fig. 5C–E), although we did consistently observe one or two sporadic, small hemorrhages in *Tgfb1^fl/fl;Pdgfrb^Cre* and *Tgfb1^fl/fl;Tie2^Cre* embryos (arrowheads in Fig. 5H and I). In areas of

hemorrhage in *Tgfb1^fl/fl;Pdgfrb^Cre* embryos, we observed no reduction in P2RY12 expression (Fig. 5H), suggesting that hemorrhage and the loss of microglial homeostasis is separable. This observation supports the model that the microglial phenotypes observed in *Itgb8* and *Tgfb1* mutant mice are due to reduced TGFβ1 activation and signaling in microglia, and not secondary consequences of hemorrhage.

**Fig. 3 | Dysmature microglia show pervasive epigenetic changes associated with MGnD and BAM gene upregulation. A** Tracks of ATAC-seq and H3K9ac ChIP-seq from *Itgb8*$^{fl/fl}$; *Emx1*$^{Cre}$ mutant and *Itgb8*$^{fl/fl}$ control microglia illustrating key homeostatic, border associated macrophage (BAM) and MgND marker gene bodies. **B** Comparison between differentially enriched genes (DEGs) (RNA-seq) and genes associated with differentially accessible peaks (DAPs) (ATAC-seq) in cortical and hippocampal microglia from *Itgb8*$^{fl/fl}$; *Emx1*$^{Cre}$ vs *Itgb8*$^{fl/fl}$ control brains (Padj < 0.05). Linear correlation between overlapping DAPs and DEGs, Log2FC. **C** Comparison between DEGs (RNA-seq) and genes associated with differentially enriched H3K9 ChIP-Seq peaks (DEPs) in *Itgb8*$^{fl/fl}$; *Emx1*$^{Cre}$ (Padj < 0.05) vs control

brains. Linear correlation between overlapping DEPs and DEGs Log2FC. **D** Heatmap of ATAC-seq showing top 100 DAPs comparing *Itgb8*$^{fl/fl}$; *Emx1*$^{Cre}$ to control microglia. *n* = 7 control, 7 mutant. **E** Heatmap of H3K9ac ChIP-seq showing top 100 DEPs comparing *Itgb8*$^{fl/fl}$; *Emx1*$^{Cre}$ to WT microglia. *n* = 4 control, 3 mutant; each biological replicate was derived from pooled microglia from 3 mice. **F** Motif enrichment (HOMER) for positive or negative enriched peaks. For epigenetic analysis, differential enrichment was determined with DESeq2 (negative binomial generalized linear model and Wald test for significance testing). Differentially accessible peaks were determined with adjustment for false discovery using benjamini hochberg method; Padj < 0.05.

---

### *Tgfb1* is required to maintain postnatal microglial homeostasis

Bulk RNA-seq analysis indicates that *Tgfb1* expression in the adult mouse brain is found primarily in microglia and endothelial cells (Fig. 6A)[48]. To assess whether there is an ongoing requirement for *Tgfb1* in postnatal microglial homeostasis we induced recombination in *Tgfb1*$^{fl/fl}$;*Cx3cr1*$^{CreER}$ (Fig. 6B–E) neonatal mice by giving tamoxifen (50uL of 10 mg/mL) on postnatal day (P)4,5,6. These mice were then collected at P30 for histological analysis. In these neonatally recombined mice, we found large patches of dysmature microglia with reduced expression of *Tgfb1* mRNA, reduced pSMAD3 staining (a measure of downstream TGFβ signaling), reactive morphology, loss of homeostatic markers TMEM119 and P2RY12, and upregulation of CD206 (Fig. 6B–E). The size and distribution of patches was highly variable, with intermixing of *Tgfb1*-intact and dysmature microglia, suggesting that *Tgfb1*-intact microglia cannot compensate for the loss of *Tgfb1* in adjacent dysmature cells. *Tgfb1* intact microglia had no reduction in pSMAD3 staining, arguing against non-autonomous effects of *Tgfb1* deletion on TGFβ signaling in neighboring microglia. Similarly, *Tgfb1*-expressing vascular cells were often found intermingled with *Tgfb1*-deleted microglia, indicating that *Tgfb1* from endothelial cells cannot compensate for the loss of *Tgfb1* in neighboring cells. *Tgfb1* tamoxifen-inducible mutants exhibited no obvious neurobehavioral abnormalities, possibly due to variable penetrance and/or an attenuated dysmature microglial phenotype.

To assess the role of *Tgfb1* in postnatal adolescent microglial homeostasis, we performed experiments to conditionally delete *Tgfb1* during this developmental window. Following tamoxifen inductions at P30, we were unable to successfully recombine the *Tgfb1* floxed allele with *Cx3cr1*$^{CreER}$, which we attributed to the large distance between the loxP sites in the *Tgfb1* floxed allele (4.4 kb)[55]. However, in mice homozygous for *P2ry12*$^{CreER}$ (*Tgfb1*$^{fl/fl}$;*P2ry12*$^{CreER/CreER}$) that were induced starting at P30 and analyzed at P60, we were able to find scattered mosaically recombined microglia that displayed a phenotype similar to neonatally induced *Tgfb1*$^{fl/fl}$;*Cx3cr1*$^{CreER}$ mice with patchy recombination (reactive morphology, loss of homeostatic markers, upregulation of CD206, and loss of pSMAD3 staining) (Fig. 6F, G). Taken together, these data indicate that microglia require self-produced TGFβ1 to maintain homeostasis.

### Reduced dysmaturity correlates with less severe neuromotor dysfunction in *Smad2/3*$^{fl/fl}$;*Cx3cr1*$^{Cre}$ mutant mice

After engaging the TGFβ receptor, TGFβ signaling proceeds through both canonical (SMAD-dependent) and non-canonical (SMAD-independent) pathways (Fig. 7A). To understand whether microglia abnormalities and associated neuromotor dysfunction can be attributed to canonical versus non-canonical TGFβ signaling, we generated *Smad2/3*$^{fl/fl}$;*Cx3cr1*$^{Cre}$ mice and compared the behavioral, transcriptional and histological properties of these mice to *Tgfb1*$^{fl/fl}$;*Cx3cr1*$^{Cre}$ mice[56,57]. Neither *Smad2/3*$^{fl/fl}$;*Cx3cr1*$^{Cre}$ or *Tgfb1*$^{fl/fl}$;*Cx3cr1*$^{Cre}$ mutants showed evidence of neonatal brain hemorrhage. Following mice into adulthood, *Tgfb1*$^{fl/fl}$;*Cx3cr1*$^{Cre}$ mice presented at 2 months of age with neuromotor dysfunction highly similar to 2-month-old *Itgb8*$^{fl/fl}$; *Nestin*$^{Cre}$, *Tgfbr2*$^{fl/fl}$;*Cx3cr1*$^{CreER16}$ and *Lrrc33*$^{-/-}$ mice[22]. In contrast, *Smad2/3*$^{fl/fl}$;*Cx3cr1*$^{Cre}$ mice showed no obvious behavioral abnormalities

until 15 months of age, at which point some mice exhibited a slight tremor (Fig. 7B). None of the *Smad2/3* mutants of the allelic series (e.g., double heterozygotes, homozygous floxed mutant of one floxed *Smad* allele with heterozygosity of the other) showed any behavioral or histological phenotype, suggesting that *Smad2* and *Smad3* can compensate for each other.

To understand why *Smad2/3* double mutants lacked neurobehavioral symptoms comparable to other *Itgb8*-*Tgfb1* signaling pathway mutants, we compared the transcriptional properties of *Itgb8*, *Lrrc33*, *Tgfb1*, *Tgfbr2* and *Smad2/3* conditional mutant models using whole brain bulk RNA-Seq (Fig. 7C–E, S6, S7 and S8). Whole brain transcriptomes from TGFβ pathway mutants were highly correlated, as would be expected from disrupting genes in a shared signaling pathway (Fig. 7C). *Lrrc33* and *Itgb8* mutants, the two most proximal to TGFβ1 activation, were the two most highly correlated gene sets. In contrast, *Smad2/3* mutants, being the most distal from TGFβ1 activation, were the least correlated, possibly reflecting the accumulation of interceding non-canonical signaling pathways. We also identified several key differences among pathway mutants. First, we noticed that the pathway genes themselves were differentially altered in the various mutants (Fig. 7D) suggesting compensatory pathway feedback; this compensation was lowest in the *Smad2/3* mutants. Beyond these potential feedback-associated gene changes, the most highly differentially expressed genes were largely microglia-specific, which we confirmed by comparing to bulk RNA-seq of sorted microglia from each model (Fig. 7E, S6). Notably, while the reduction in homeostatic genes in *Smad2/3* mutant mice was comparable to other models, increases in MGnD/DAM and BAM genes were generally less pronounced in *Smad2/3* mutants than in all other mutants (Fig. 7E). Additionally, while *Tgfb1*$^{fl/fl}$;*Cx3cr1*$^{Cre}$ mice showed significant upregulation of the astrocytosis-related proteins GFAP and VIM, genes associated with astrocytosis were not as highly upregulated in *Smad2/3* mutants as compared to other TGFβ mutant models (*ApoE*, *GFAP*, *Fabp7*, and *Vim*) in our bulk RNA-seq analysis (Figure S7). Focusing on genes that were differentially expressed in *Tgfb1-* and *Tgfbr2-*, but not *Smad2/3*-mutants yielded several putative biological processes enriched in symptomatic mutants. Myelination and complement associated synaptic pruning were among the most representative terms associated with *Tgfb1*$^{fl/fl}$;*Cx3cr1*$^{Cre}$ and *Tgfbr2*$^{fl/fl}$;*Cx3cr1*$^{Cre}$ microglia and whole brain RNA-seq gene sets, which also differentiate these models from *Smad2/3*$^{fl/fl}$;*Cx3cr1*$^{Cre}$ mice (Figure S8). Bioinformatic analysis of sex-specific microglial phenotypes did not reveal a sex-dependent microglial signature in the TGFβ mutant models that we analyzed, nor did a histological analysis of adult male and female *Lrrc33*$^{+/-}$ control and *Lrrc33*$^{-/-}$ mutant microglia (Figure S9).

Among microglia-specific genes differentially expressed in *Smad2/3*$^{fl/fl}$;*Cx3cr1*$^{Cre}$ versus other mutant models, we found *Apoe*, *Clec7a* and *Lgals3* to be of particular interest, given their link to disease states (Fig. 7E). While considered canonical markers of DAM/MGnD microglia, these genes are also transiently enriched in populations of developmentally restricted microglia associated with myelin and axonal tracts (axonal tract microglia, ATM)[58], and in areas of active neurogenesis (proliferation associated microglia, PAM)[59]. We previously found evidence for delayed myelination in *Itgb8*$^{fl/fl}$;*Nestin*$^{Cre}$ and

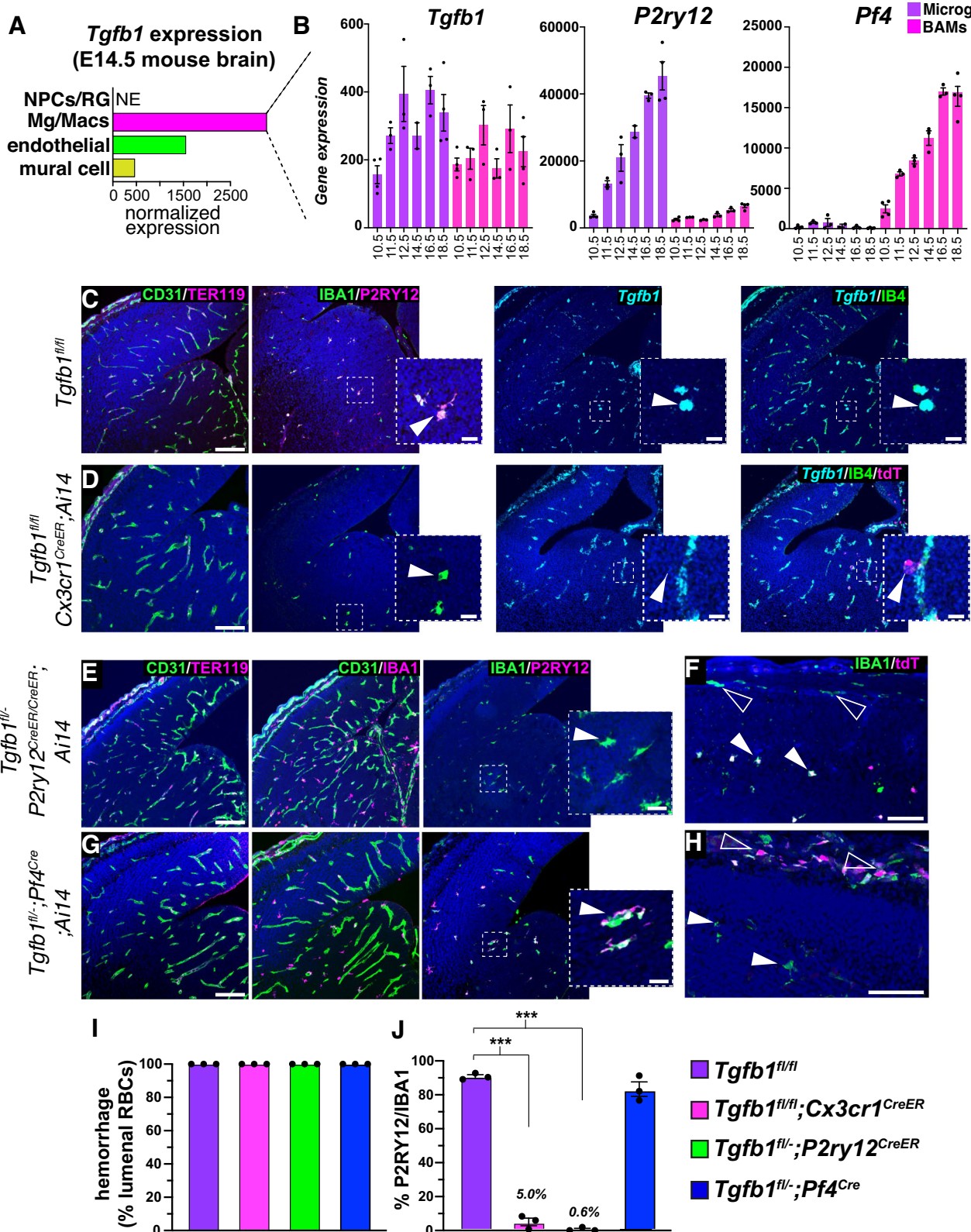

Tgfbr2[fl/fl];Cx3cr1[Cre] mutant mice[16]. The expression of these genes in MGnD microglia depend on the APOE pathway: MGnD signature genes including *Spp1*, *Gpnmb*, and *Clec7a* are suppressed in disease models that also lack *Apoe*[11,12], and expression of human APOE isoforms APOE ε4 in microglia similarly suppresses MGnD polarization relative to the ε3 isoform[39]. In contrast, developmental expression of MGnD/ATM genes near myelin/axonal tracts does not depend on *Apoe*[59].

We were therefore interested to understand the degree to which *Apoe* might regulate PAM/MGnD signature genes, and whether increased expression of APOE in *Itgb8*, *Lrrc33*, *Tgfb1*, and *Tgfbr2* mutants might account for their more severe neuromotor symptoms. To that end, we performed an epistatic knockout analysis by deleting *Tgfbr2* or *Apoe* singly or together. We created *Tgfbr2[fl/fl];Cx3cr1[Cre];Apoe[+/−]* and *Tgfbr2[fl/fl];Cx3cr1[Cre];Apoe[−/−]* mutant mice, and evaluated their

**Fig. 4 | Microglia provide their own TGFβ1 to promote and maintain home-
ostasis. A** Sorted bulk-Seq analysis[33] of embryonic microglia and BAMs reveals that
*Tgfb1* is expressed in both microglia and BAMs during embryonic development,
whereas (**B**) *P2ry12* and *Pf4* are specific markers of these two respective popula-
tions. **C**, **D** Analysis of control (**C**) and conditional *Cx3cr1^{CreER}* mediated deletion (**D**)
of *Tgfb1* deletion in the E14.5 forebrain following E11.5, 12.5 and 13.5 tamoxifen
induction. Analysis revealed no hemorrhage (CD31 in green, TER119 in magenta), no
change in macrophage/blood vessel association (IBA1 in magenta, CD31 in green),
loss of the homeostatic marker P2RY12 (in magenta, IBA1 in green), and loss of
*Tgfb1* (cyan) in Isolectin B4 (green) and tdT (red) labeled microglia, but not in IB4⁺
labeled blood vessels. **E** Analysis of conditional *P2ry12^{CreER}* mediated deletion of
*Tgfb1* deletion in E14.5 microglia following E11.5, 12.5 and 13.5 tamoxifen induction.
Analysis revealed no brain hemorrhage (CD31 in green, TER119 in magenta), no
change in macrophage/blood vessel association (IBA1 in magenta, CD31 in green),
and loss of the homeostatic marker P2RY12 (magenta), in IBA1⁺ (green) microglia.

**F** *P2ry12^{CreER}* recombination, as shown by *ROSA-tdTomato (Ai14)* Cre reporter
expression, was restricted to microglia (closed arrowheads), and was not seen in
the overlying meninges (open arrowheads). **G** Analysis of *Pf4^{Cre}* mediated deletion
of *Tgfb1* in the E14.5 forebrain. Analysis revealed no hemorrhage (CD31 in green,
TER119 in magenta), no change in macrophage/blood vessel association (IBA1 in
magenta, CD31 in green), and no loss of the homeostatic marker P2RY12 (magenta),
in IBA1⁺ (green) microglia. **H** *Pf4^{Cre}* recombination was restricted to the embryonic
meninges (open arrowheads), and was not seen in microglia (closed arrowheads). **I**, **J** Quantification of hemorrhage (**I**) and microglial P2RY12 expression (**J**) in dif-
ferent Tgfb1 conditional mutant models. *n* = 3 for histological analysis. In 2B, *n* = 2
(E14.5), *n* = 3 (E11.5, 12.5, 16.5) and *n* = 4 (E10.5, E18.5). ***$p < 0.001$, two-tailed t-test.
In **J**, $p = 4.3 \times 10^{-6}$ and $2.28 \times 10^{-7}$ for *Cx3cr1^{CreER}* and *P2ry12^{CreER}* comparisons to control,
respectively. Error bars= mean +/- SEM. Gene expression in **B** = normalized RPKM.
Scale bar in **B–E** and **G** = 150$\mu$m, inset scale bar=20$\mu$m. Scale bar in **F** and **H** = 50$\mu$m.
Source data are provided as a Source Data file.

---

behavioral and microglial phenotypes at P90[60] (Figure S10). Surpris-
ingly, despite the transcriptomic similarity between dysmature and
MGnD microglia, we found that deletion of APOE in *Tgfbr2^{fl/fl};Cx3cr1^{Cre}*
mice had no major or obvious effects on neuromotor or microglial
phenotypes (Figure S10), although we did not analyze these mice for
subtle changes in neuromotor behavior or gene expression. This is
similar to PAM/ATM microglia which also do not depend on APOE[59].
This genetic epistatic analysis indicates that, unlike MGnD microglia,
dysmature microglia do not depend on APOE for their polarization to
occur, or alternatively, that non-canonical TGFβ signaling functions
directly downstream of APOE in microglial polarization.

To determine the underlying causes of the motor dysfunction
seen in TGFβ pathway mutants, we considered the non-autonomous
roles of disease-associated molecules secreted by dysmature micro-
glia. In Alzheimer's disease mouse models, astrocyte activation is
correlated with the degree of MGnD microglial polarization, and is
mechanistically linked to microglia expression of LGALS3[39]. Examina-
tion of LGALS3 expression by immunohistochemistry and whole brain
western blot analysis confirmed that LGALS3 expression was much
higher in *Tgfb1^{fl/fl};Cx3cr1^{Cre}* than *Smad2/3^{fl/fl};Cx3cr1^{Cre}* mice, providing a
potential explanation for different behavioral outcomes in these mice,
despite the loss of microglial homeostatic markers in both models
(Fig. 7F–H, Figure S11), and in line with our observations regarding
astrocytosis-related gene expression (Figure S7). While most brain
regions in *Smad2/3^{fl/fl};Cx3cr1^{Cre}* mice did not show microglial LGALS3
expression, areas containing dense white matter, such as the corpus
callosum, showed upregulation of LGALS3 (asterisk in Fig. 7H). How-
ever, this observed upregulation in the white matter of *Smad2/3^{fl/fl};
Cx3cr1^{Cre}* did not lead to statistically significant differences in LGALS3
expression in *Smad2/3^{fl/fl};Cx3cr1^{Cre}* mice as compared to *Tgfb1^{fl/fl};
Cx3cr1^{Cre}* in whole brain western blot analysis (Figure S11). *Tgfb1^{fl/fl};
Cx3cr1^{Cre}* had much higher relative levels of microglial LGALS3 upre-
gulation (arrowheads in Figs. 7G and 7H) in the corpus callosum and
fiber tracts of the striatum (arrowheads in Figs. 7G, 3.40+/−0.11-fold
increase by whole brain western blot, Figure S11), suggesting that white
matter microglia may be particularly sensitive to the effects of TGFβ
signaling disruption. Together, these data indicate that isolated dis-
ruption of canonical (SMAD-directed) TGFβ signaling in microglia
results in neuropathological phenotypes that are less severe than in
mice with more proximal disruptions in the TGFβ signaling cascade,
and that the disease-associated microglial gene LGALS3 may drive
astrocytic changes seen in *Tgfb1^{fl/fl};Cx3cr1^{Cre}* mice that are less severe in
*Smad2/3^{fl/fl};Cx3cr1^{Cre}* mice (Fig. 7 and S7).

## Discussion
Here, we show that that microglial differentiation is dependent upon
the activation of microglial TGFβ1 by *Itgb8*-expressing radial glia pro-
genitors. Disruption of this signaling by conditionally deleting radial

glia progenitor *Itgb8*, microglial *Tgfb1*, or the downstream transcrip-
tion factors *Smad2/3* in microglia results in a microglial transcriptional
phenotype characterized by the absence of mature microglial markers
and persistent expression of genes normally expressed in microglial
precursors and immature microglia.

We find that domain-specific deletion of *Itgb8* in early embryonic
radial glia progenitors results in microglial defects that are restricted
to the brain regions created by these progenitors. Furthermore, con-
sistent with recent reports that microglia are derived from a CD206⁺
precursor[34–36] and the earliest characterizations of microgliogenesis by
del Rio-Hortega[37], we find that embryonic radial glial endfeet make
direct contact with CD206⁺ meningeal macrophages. Together, our
data support the model that physical interactions between radial
progenitors and immature microglia are necessary for *Itgb8*-mediated
TGFβ signaling. This is consistent with previous reports of microglia/
radial glia interactions[61]. The lack of a similar microglial phenotype in
*Itgb8^{fl/fl};hGFAP^{Cre}* mutants, which recombine dorso-lateral radial pro-
genitors ~3 and ~5 days after *Nestin^{Cre}* and *Emx1^{Cre}* respectively,
places the timing of this interaction between E9 and E13.5, when microglia
initially begin to invade the developing nervous system[27,47]. Our data
indicate that dysmature microglia in *Itgb8/Tgfb1* pathway mutants are
quite similar to myeloid progenitors at this age. Our data support the
model that microglia are derived from a CD206⁺ precursor and that
disruption of ITGB8/TGFβ1 signaling blocks microglial marker
expression and allows for maintained expression of immature markers
that would otherwise be downregulated. Our analysis shows that this is
reflected by widespread epigenetic differences in homeostatic, MGnD/
DAM and BAM gene loci. Additional future studies are needed to
determine where and more precisely when microglial/radial glia
interaction occurs developmentally, and also whether the specific
interactions between ingressing microglia and radial glial basal and/or
apical endfeet in the meninges, on brain blood vessels, or at the ven-
tricular surface are necessary for microglial differentiation.

Our findings suggest that additional new models of brain region
restricted microglial dysfunction can be created by deleting *Itgb8* in
other radial progenitor domains. Taking advantage of the integrin $\alpha_V\beta_8$
trans-activation of TGFβ1 to create genetic models of brain region-
restricted microgliopathy is particularly useful, as microglial *Cre* and
*CreER* lines generally recombine microglia throughout the nervous
system, and not in particular brain regions[36,50,62,63]. We believe that the
study of brain region-restricted microglial dysfunction will provide
crucial information about how dysmature microglia in particular brain
regions contribute in a modular fashion to more complex behavioral
and cognitive disruptions seen in other animal models and in humans
with pervasive neurodevelopmental immune dysfunction. *Itgb8^{fl/
fl};Emx1^{Cre}* mice do not display the significant motor deficits seen in
*Itgb8^{fl/fl};Nestin^{Cre}* mice, suggesting that developmental neuroimmune
disruption of the motor and somatosensory cortex is not sufficient to

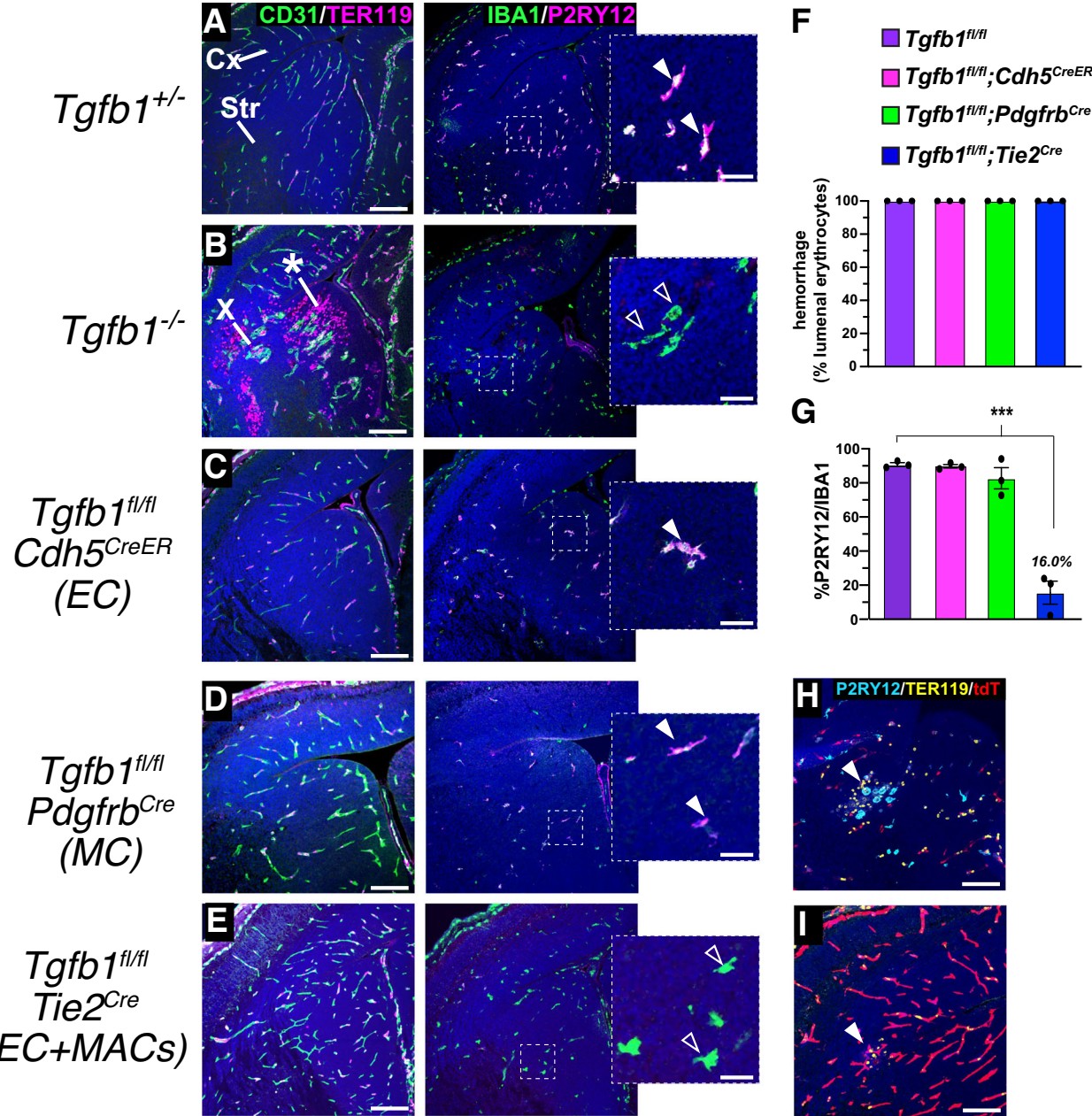

**Fig. 5 | Vascular *Tgfb1* is not required for microglial development.** E14.5 coronal brain sections from (**A**) control (*Tgfb+/-*) embryos, (**B**) embryos with global (*Tgfb1-/-*) or cell-lineage specific deletion of *Tgfb1* (*Tgfb1fl/fl*) in (**C**) endothelial cells (*Cdh5CreER*) (**D**, **H**) vascular mural cells (*PdgfrbCre*), or (**E**, **I**) endothelial cells and microglia/macrophages (*Tie2Cre*). Sections were stained for hemorrhage (TER119, magenta) and vasculature (CD31, green) or for committed/hemostatic microglia (IBA1, magenta and P2RY12, green), or to study P2RY12 expression in the context of hemorrhage (P2RY12, cyan and TER119, yellow). Only *Tgfb1-/-* mutants have consistent evidence of vascular dysplasia (marked by X) and hemorrhage (asterisk), whereas mice with microglia/macrophage deletion of *Tgfb1* (*Tgfb-/-*, and *Tgfb1fl/fl*;

*Tie2Cre* mutants) have presence of dysmature microglia (open arrowheads, blowups to right). Quantification of hemorrhage (**F**) and microglial P2RY12 expression (**G**) in different *Tgfb1* conditional mutant models. ***$p$ = 0.0004 in **G**. Panels in **H** and **I** show *ROSA-tdTomato (Ai14)* recombination pattern of *PdgfrbCre* and *Tie2Cre* mouse lines respectively, and P2RY12 expression (or lack thereof) in the sporadic hemorrhage (arrowheads) seen when *Tgfb1* is deleted with these lines. $n$ = 3. ***$p$ < 0.001, two-tailed t-test. Error bars= mean +/- SEM. Cx=Cortex, Str=Striatum. Scale bar in (**A**–**E**, **H** and **I**) = 150$\mu$m, inset scale bar=30$\mu$m. Source data are provided as a Source Data file.

disrupt motor function in mice. Future work using other spatially restricted *Cre* and *CreER* lines should provide localization of the neuroimmune dysfunction underlying the complex neurobehavioral changes observed in *Itgb8fl/fl;NestinCre* mice. Combining mouse models of neuroinflammatory or neurodegenerative disease with brain region-specific *Itgb8* conditional knock-out models will be especially illuminating.

In line with recent cryoEM data[19,20], we demonstrate that microglia depend on cell-autonomous expression of *Tgfb1* for their early

maturation. We only observe a fulminant vascular/hemorrhage phenotype in *Tgfb1* null mice; deletion of *Tgfb1* in endothelial cells and microglia (*Tie2Cre*), or in vascular mural cells and fibroblasts (*PdgfrbCre*) had no major effect on cerebrovascular morphogenesis or barrier function. Therefore, vessel-expressed TGFβ1 is apparently dispensable for microglial maturation, but the source of TGFβ1 for vascular maturation remains unknown. Fitting with the paracrine activation / autocrine signaling model, we propose that radial glia ITGB8 activates TGFβ1 on endothelial cells and/or pericytes, so that the loss of

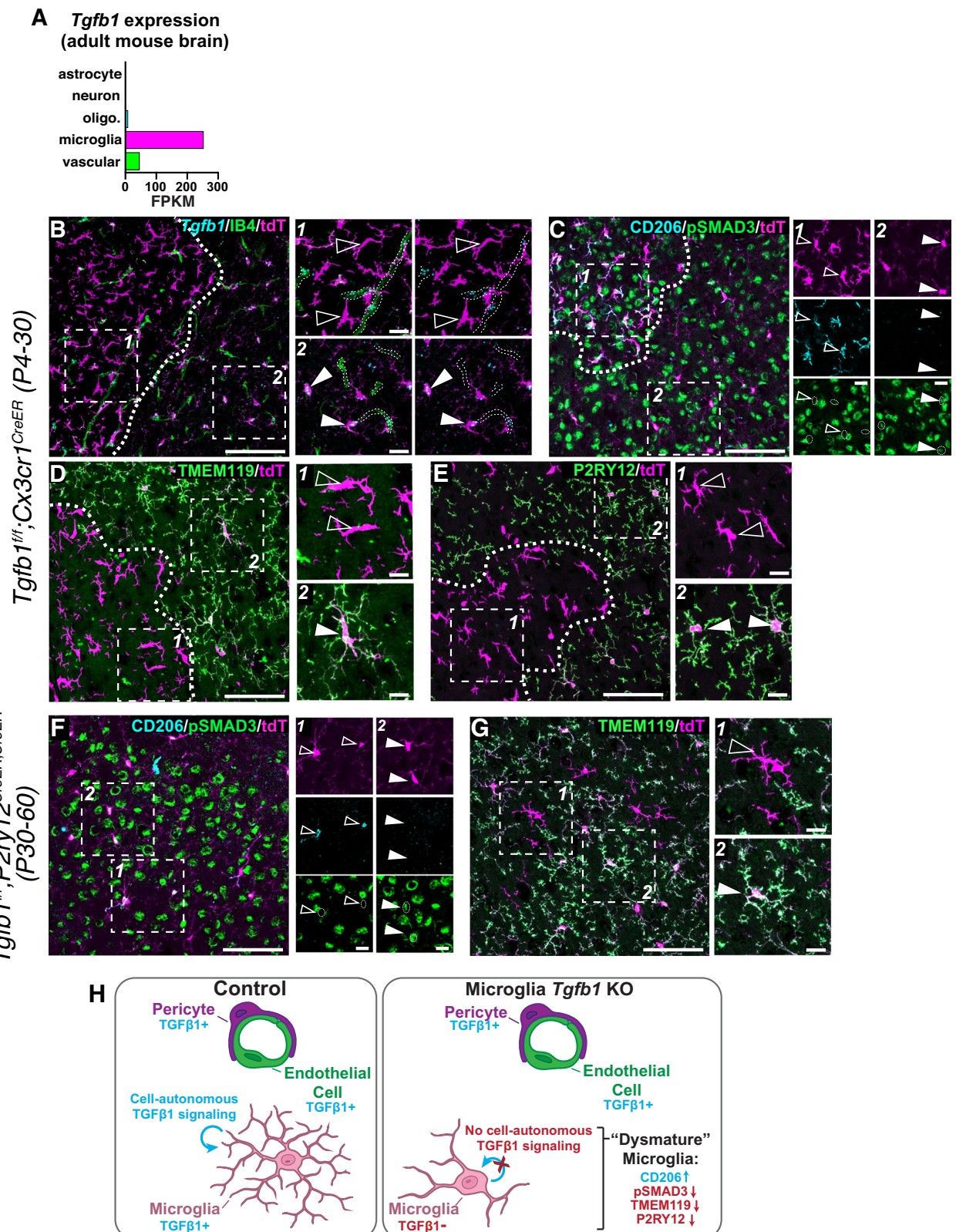

endothelial TGFβ1 can be compensated by pericyte TGFβ1 or vice versa. It is also possible that TGFβ1 in the embryonic circulation can compensate for the loss of vascular TGFβ1[54].

In contrast to *Itgb8*, which we show is largely dispensable for maintaining microglial homeostasis, we find that microglial *Tgfb1* is necessary for microglial homeostasis postnatally. One potential explanation for this discrepancy could be the activation of TGFβ1 by

other integrin complexes or by integrin-independent pathways postnatally. While our manuscript was under review, another group also reported findings regarding the role of TGFβ1 in the regulation of microglial homeostasis in adulthood, in line with our findings[64]. We recently found that antibody-based blockade of ITGB8 in AD mouse models enhances MGnD polarization and reduces the size of amyloid plaques[39]. Conditional deletion of *Lrrc33* from microglia/macrophages

**Fig. 6 | *Tgfb1* is required postnatally for microglial homeostasis. A** Bulk-seq analysis of *Tgfb1* expression in the adult mouse brain[48]. Analysis revealed enrichment for *Tgfb1* expression primarily in microglia and vascular cells. **B**–**E** Analysis of conditional *Cx3cr1^CreER* mediated deletion of *Tgfb1* deletion in the P30 mouse cerebral cortex following neonatal tamoxifen induction at P4,5 and 6. Analysis revealed a patchy distribution of dysmature microglia characterized by altered morphology and (**B**) *Tgfb1* loss, (**C**) upregulation of CD206 and loss of pSMAD3 staining, (**D**) loss of the homeostatic marker TMEM119 and (**E**) loss of the homeostatic marker P2RY12. Open arrowheads in B-G mark dysmature microglia, closed arrowheads mark dysmature microglia. Postnatal deletion of *Tgfb1* (P30-P60) using *P2ry12^CreER* resulted in isolated production of microglia with altered morphology (**F**), loss of pSMAD3 staining (**F**), downregulation of the homeostatic marker TMEM119 (**G**), and upregulation of CD206 (**F**). **H** Cartoon depiction of the role of TGFβ1 in the cell-autonomous control of microglial homeostasis. Microglial cartoon created in BioRender. McKinsey, **G.** (2025) https://BioRender.com/r69o433. Scale bar in (**B**–**G**) = 100μm, inset scale bar=20μm.

using *Cx3cr1^CreER* resulted in no major transcriptional or behavioral phenotypes compared to mice with global or conditional deletion of *Lrrc33* during development, which virtually phenocopy *Itgb8^fl/fl;Emx1^Cre* and *Itgb8^fl/fl;Nestin^Cre* mutants[22]. This raises the possibility that proteins that present/modify *Tgfb1* are only active developmentally or in disease contexts, an important consideration for the design of TGFβ-targeting therapeutics.

In line with previous reports[17,42,58,65], our study emphasizes the common features of developmental and disease-associated microglia. In particular, by comparing *Smad2/3* conditional mutants to other conditional TGFβ mutant models, we find that there is a phenotypic gradient characterized by lower vs higher relative expression of MGnD and BAM genes and directly correlated neuromotor dysfunction. We propose that polarization along this gradient is regulated by non-canonical TGFβ signaling, a model that is supported by previous observations that disruption of non-canonical TGFβ signaling (e.g., TGFβ-associated kinase, TAK1-signaling) promotes microglial homeostasis[66,67]. NFKB, a major downstream signaling target of TAK1 regulation[68], is a master regulator of neuroinflammatory responses and has been implicated in the etiology of neurodegenerative disease[69–71]. Our findings are also consistent with a recent study of the role of *Smad4* in microglial differentiation[72]. More direct comparisons between these models are needed to explore the molecular and neurobehavioral effects of *Smad2/3* and *Smad4* conditional mutation, and how non-canonical TGFβ signaling might amplify or suppress these effects. Importantly, we find that while the dysmature gene signature may be protective in neurodegenerative disease, when it is present during development, mice acquire severe neurological impairments. Additional studies are needed to better understand how microglial reactivity is regulated in different brain regions during normal development and how these same gene regulatory pathways manifest in the context of brain injury and disease.

## Methods

### Mice
We used Mus musculus in this study. All mouse studies were followed by the protocol approved by the Institutional Animal Care and Use Committee (IACUC) at UCSF, protocol #AN194997. All mice were backcrossed to C57BL6/J, except for *Tgfb1* constitutive null mice, which were on the NIH/OlaHsd background. Mice were separated by sex and housed on a 12:12 light:day cycle, with no more than 5 mice per cage. Mouse room temperature was set to a range of 67 to 74 °F and relative humidity was held to a range of 30-70%. Mice were given food (PicoLab Rodent Diet 20 #5053, LabDiet) and water *ad libitum*. Mice not collected for experimental analysis were euthanized using CO2 inhalation, followed by cervical dislocation. For all quantification and statistical comparisons, biological replicates were derived from separate mice, with an n of at least 3 for each category. Findings applied to both sexes. Sex-dependent differences were not a primary variable of interest, with the exception of the histological analysis of *Lrrc33* mutants. Mutant mice were generated by crossing a Cre+; floxed allele (f/+) or null allele (+/−) parents to a homozygous flox (F/F) parent. We designed our breeding strategy based on recommendations Genetic Resource Science group at The Jackson Laboratory and published by Luo et al.[73]. All WT, floxed, and recombined alleles were assessed by PCR genotyping. We note evidence of parental sex bias in germline recombination in *hGFAP^Cre* mice. With our breeding strategy, there were no true wild type mice (without transgenes or floxed alleles). Rather, comparisons were generally made between fl/+;Cre (control) and fl/fl;Cre (mutant). Please see key resources table for additional mouse line information. All tdT reporter analysis was done with the *ROSA-tdTomato (Ai14)* mouse line, except for the embryonic *Nestin^Cre* mice in Fig. 1 and *Syn1^Cre* mice in Figure S4, which were crossed to the membranous tomato reporter expressing Cre-dependent *iSuRe-Cre* mouse line.

### Histology and Immunostaining
Prior to harvest, mice were given an overdose of saturated isofluorane via inhalation, followed by bilateral thoracotomy, according to UCSF IACUC guidelines. Postnatal mouse brains were harvested at P30 or P60-P90 following transcardial perfusion with 20 mL cold PBS and 20 mL cold 4% formaldehyde. Brains and E14.5 embryos were fixed overnight at 4 degrees in 4% formaldehyde, followed by overnight incubation in 30% sucrose. Samples were embedded (Tissue Plus O.C.T. Compound, Fisher Scientific, 23-730-571) and sectioned at 20 μm. Some adult brains were sectioned at 40um. Sections were immunostained using a blocking/permeabilization buffer of PBS containing 2% BSA, 5% donkey serum (Sigma, D9663) and. 5% TritonX-100. Primary and secondary antibodies were diluted in PBS containing 1% BSA and. 25% TritonX-100. Secondary antibodies conjugated to Alexa fluorophores were used at 1:300 (Jackson ImmunoResearch). Immunostained sections were mounted with Prolong Gold Antifade Mountant (Thermofisher, P36930). Please see Table 1 for list of antibodies and the relevant concentrations used for immunofluorescence.

### Western blot
Isolated protein samples (10μg and 40μg for ITGB8 and LGALS3 blot respectively) were resolved in 10% SDS-PAGE and transferred to polyvinylidene difluoride (PVDF) membranes (Millipore, IPVH08100). The membranes were incubated for blocking in 5% bovine serum albumin (BSA)–Tris-buffered saline containing Tween (TBST) for 1 h at room temperature. The membranes were separately incubated with primary antibodies against ITGB8 (gift from Dr. Joseph McCarty), LGALS3 (Cedarlane, CL8942AP), GAPDH (R & D Systems, MAB5718), beta actin (R & D Systems, MAB8929), diluted in 5% BSA–TBST overnight at 4 °C. The membranes were incubated with secondary antibodies conjugated with Horse radish peroxidase (HRP) diluted in 5% BSA–TBST followed by three washes with 1X TBST. The bands were detected using a LiCOR chemiluminescence detector and subsequent densitometric quantifications were performed by using Fiji (ImageJ) software and Graphpad Prism 9. Each blot was normalized with its respective loading control before calculation of the fold changes.

### Neurobehavioral analyses
Hind limb stride length was measured by the method adapted by Zhang et al.[74]: Hind paws of mice were wetted with ink. Animals were then placed on a strip of 3MM filter paper (4.5 cm wide, 40 cm long). Stride lengths were measured as the distance between two hind paw prints. For rotorod analyzes, mice were placed on a 6-cm diameter rod A Rotarod device (Ugo Basile) accelerated from 0 to 40 rpm over

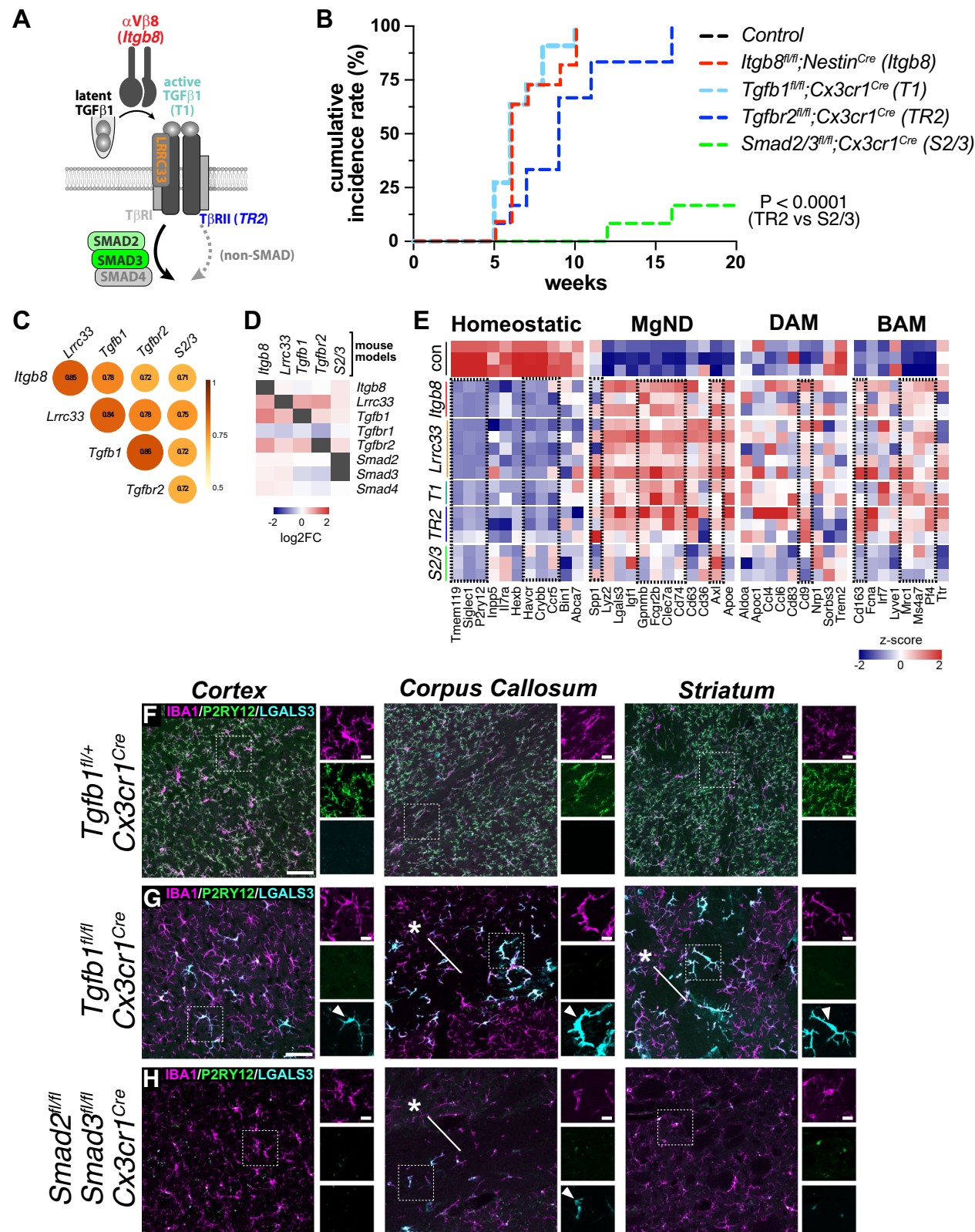

5 min. Mice received three 5-min trials with a 2-h intertrial interval. The amount of time spent on the rod before falling off was measured. For observational outcome measures, the following scoring system was adapted from Wang et al.[75]. For "Gait": A score for walking gait was given using a three-point observational scoring system. "2" indicated normal gait; "1" indicated broad-based hind limbs and waddling while walking; "0" indicated severe abnormalities with inability to ambulate,

dragging limbs, severe tremor. "Appearance": A score for appearance of general condition was given using a three-point observational scoring system to assess coat condition, appearance of eyes, and body stance. "2" indicated clean shiny coat, clear eyes, and normal stance; "1" indicated dull coat, ungroomed appearance, dull eyes, and hunched stance; "0" indicated piloerection, crusted eyes, and kyphosis. "Tremor": A score was given for tremor or spastic movements using a

**Fig. 7 | Disruption of non-canonical TGFβ signaling in microglia drives disease-associated gene expression. A** Schematic of TGFβ signaling, with analyzed mutant mouse models analyzed by bulk and microglial flow cytometry noted by color (*Itgb8*=red; *Tgfb1*=cyan; *Lrrc33*=orange; *Tgfb2*=blue, *Smad2/3*=green). **B** Cumulative incidence of motor dysfunction seen in different conditional TGFβ pathway mutants as a function of age. Incidence included any sign of detriment to gait, appearance, or tremor, based on a 0,1,2 rating scale (see[16] for details). *p* < 0.0001, two-tailed t-test. **C** Pearson´s correlation coefficient of bulk-seq normalized expression of differentially expressed genes across TGFβ mutant models.
**D** Compensatory transcriptional changes of key TGFβ signaling genes in different TGFβ mutant models. Heatmap shows log$_2$FC. **E** Bulk-seq analysis of microglial

homeostatic and disease associated (MGnD/DAM) microglial markers across TGFβ mutant models. Heatmap shows z-score of normalized expression. Differentially expressed genes in all mutants (*p* < 0.05; DESeq2) are indicated with dotted rectangles. **F–H** Comparison of control (**F**) *Tgfb1$^{fl/+}$;Cx3cr1$^{Cre}$*, (**G**) *Tgfb1$^{fl/fl}$;Cx3cr1$^{Cre}$* and (**H**) *Smad2/3$^{fl/fl}$;Cx3cr1$^{Cre}$* adult mice. Analysis revealed loss of the homeostatic marker P2RY12 in both conditional *Tgfb1* and *Smad2/3* mutants, and upregulation of the MGnD-associated microglial marker LGALS3 (see arrowheads). LGALS3 upregulation in *Tgfb1* conditional mutants was significantly higher in white matter (asterisks in **G** and (**H**) and was only seen in the white matter of *Smad2/3* conditional mutants (**H**). Scale bar in **F–H** = 50µm, inset scale bar=10µm.

three-point observational scoring system. "0" indicated no tremor; "1" indicated mild intermittent tremor, made worse when feet were lifted; "2" indicated constant tremor and uncontrollable spastic movements. Cumulative incidence scoring (Fig. 7b) was performed by noting on a weekly basis whether mice reached a score of 1 for any gait, appearance, or tremor, based on the 0,1,2 rating scale described above.

### RNAscope
20um cryosectioned tissue sections were processed for RNAscope using the manufactures protocol for cryosectioned tissue and reagents from the RNAscope Multiplex Fluorescent Reagent Kit v2 (ACD, 323100) and Opal secondaries (Akoya Biosciences, FP1488001KT). A custom RNAscope probe set was used to target the floxed exon 1 of *Tgfb1* (ACD, 1207831-C1). Slides were hybridized in the ACD HybEZ II Oven (ACD, 321711).

### Tamoxifen induction
Recombination was induced by three doses of tamoxifen dissolved in corn oil, administered by oral gavage every other day (150 µL of 20 mg/mL) (Sigma, T5648). For embryonic mouse inductions, pregnant dams were given tamoxifen (150 µL of 20 mg/mL) on E11.5, E12.5 and E13.5, for a total of three gavage injections. Neonatal tamoxifen injections were done at P4,5, and 6, at a dosage of 500µg injected intraperitonealy (50uL of a 10 mg/mL solution in corn oil). All mouse work was performed in accordance with UCSF Institutional Animal Care and Use Committee protocols. Mice had food and water ad libitum.

### Flow cytometry
The *Itgb8$^{fl/fl}$;Emx1$^{Cre}$* Mice were euthanized in a $CO_2$ chamber and then transcardially perfused with 10 ml cold Hanks' Balanced Salt Solution (HBSS, ThermoFisher, 14175103). The mouse brain was isolated and the cortex was dissected from one hemisphere for microglia purification using the standard isolation procedure established in Butovsky lab[76]. Briefly, the cortex was homogenized and resuspended with 5 ml 70% Percoll Plus (GE Healthcare, 17-5445-02) and 5 ml 37% Percoll Plus placed on top. The microglia were enriched in the interface layer after centrifugation in 800 g, 4 °C, for 25 min with an acceleration of 2 and a deceleration of 1. The microglia enriched cell population were stained with PE-Cy7 anti-mouse CD11b (1:300, eBioscience, 50-154-54), APC anti-mouse Fcrls (1:1000, clone 4G11, Butovsky Lab), and PerCP/Cy5.5 anti-mouse Ly-6C (1:300, Biolegend, 128012). The cells were then processed by a BD FACSAria™ II (BD Bioscience) and CD11b$^+$Fcrls$^+$ Ly-6C$^-$ cells were sorted into Eppendorf tubes for RNA-seq. Microglia from *Smad2/3$^{fl/fl}$;Cx3cr1$^{Cre}$* and *Tgfb1$^{fl/fl}$; Cx3cr1$^{Cre}$* mice were isolated using a percol gradient isolation strategy. To isolate single cells, brains were cut into small pieces and passed through a 40 µm filter. Single cell suspensions were prepared and centrifuged over a 30%/70% discontinuous Percoll gradient (GE Healthcare) and mononuclear cells were isolated from the interface. Flow cytometry was performed on a FACS Aria

III using the FACSDiva 8.0 software (BD Biosciences) and was analyzed using FlowJo v10.6.1. TdTomato+ cells were analyzed for CD45 and CD11b expression following initial gatings for forward scatter, singlets and live cells.

### RNA sequencing
For the sequencing of microglia isolated from *Itgb8$^{fl/fl}$;Emx1$^{Cre}$* mice, one thousand sorted microglia resuspended in 5 ml TCL buffer with 1% 2-Mercaptoethanol (Millipore-Sigma, M26250-10mL) were placed into a 96-well plate and sent to Broad Technology Labs for Smart-Seq2 following their standard protocol. Briefly, cDNA libraries were generated using the Smart-seq2 protocol[77]. The amplified and barcoded cDNA libraries were loaded into Illumina NextSeq500 (Illumina SY-415-1001) sequencer using a High Output v2 kit (Illumina FC-404-1005) to generate 2 × 25 bp reads. For whole brain samples, RNA isolation was performed using dounce homogenization of dissected cerebral cortex samples, followed by Trizol (Thermofisher, 15596026) extraction and alcohol precipitation. For Total RNA from FACS-sorted microglia from *Smad2/3$^{fl/fl}$;Cx3cr1$^{Cre}$* and *Tgfb1$^{fl/fl}$;Cx3cr1$^{Cre}$* mice, RNA was isolated using the QIAGEN RNAeasy micro kit. PolyA$^+$ unstranded libraries were synthesized from total RNA (RIN > 6) with NEBNext Ultra II RNA Library Prep Kit (New England Biolabs, E7770S), and sequenced in a Illumina HiSeq4000 system (150 pb paired-end setting) (Illumina, SY401-4001). Demultiplexed fastq files were aligned to the mouse genome (mm10) with Rsubread and quantified with FeatureCounts. Count normalization and differential expression analysis were performed with DESeq2, considering an FDR < 0.05. Correlations between log$_2$FC of genes in multiple datasets were computed using Pearson´s correlation with the cor() function from base R and visualized using the corrplot package. For correlations, we selected genes with differential expression in at least one dataset for a final set of 4406 genes contributing to correlations. Heatmaps were generated with the pheatmap package; units used in each heatmap are indicated in their legend. Venn diagrams were produced with standard GNU coreutils and drawn in Inkscape. Over-representation analysis was performed using PANTHER from the GO portal. Datasets used in this study are as follows: newly generated data (whole brain data, isolated *Tgfb1* mutant microglia datasets) have been deposited into the GEO database under accession GSE236615. *Itgb8$^{fl/fl}$;Emx1$^{Cre}$* microglia and *Smad2/3$^{fl/fl}$; Cx3cr1$^{Cre}$* data were generated previously by our group and are available under GEO GSE239603. *Tgfbr2$^{fl/fl}$;Cx3cr1$^{Cre}$* microglia is from GEO GSE124868. Data from *Lrrc33$^{-/-}$* microglia and whole brain was obtained from GEO GSE96938. ATAC-seq data used for comparisons with our data were obtained from GEO GSE79816.

For bulk RNA samples, RNA samples were quantified using Qubit 2.0 Fluorometer (ThermoFisher, Q32866) and RNA integrity was checked using TapeStation (Agilent Technologies, G2991BA). The RNA sequencing libraries were prepared using the NEBNext Ultra II RNA Library Prep Kit for Illumina using manufacturer's instructions (New

**Table 1 | Mouse Line and Reagent List**

| Reagent type | Designation | Source of references | Identifiers | Additional information |
|---|---|---|---|---|
| Mouse line | *Ai14* | JAX | Stock No: 007914. RRID:IMSR_JAX:007914. | |
| Mouse line | *Aldh1L1$^{CreER}$* | JAX | Stock No: 029655. RRID:IMSR_JAX:029655. | |
| Mouse line | *Apoe null* | JAX | Stock No: :002052. RRID:IMSR_JAX::002052. | |
| Mouse line | *CAG-Sun1/sfGFP* | JAX | Stock# 030952. RRID: IMSR_JAX:030952. | |
| Mouse line | *Cx3cr1$^{Cre}$* | JAX | Stock No: 025524. RRID:IMSR_JAX:025524. | |
| Mouse line | *Cx3cr1$^{CreER}$* | JAX | Stock No: 020940. RRID:IMSR_JAX:020940. | |
| Mouse line | *Emx1$^{Cre}$* | JAX | Stock No: 005628. RRID:IMSR_JAX:005628. | |
| Mouse line | *hGFAP$^{Cre}$* | JAX | Stock No: 004600. RRID:IMSR_JAX:004600. | |
| Mouse line | *iSure$^{Cre}$* | (Fernández-Chacón et al. 2019) | MGI:6361145. | |
| Mouse line | *Itgb8 flox* | (Proctor et al. 2005) | MGI: 3608910. | |
| Mouse line | *Itgb8$^{tdT}$* | (Nakawesi et al. 2021) | Obtained from Dr. Helena Paidassi. | |
| Mouse line | *Nestin$^{Cre}$* | JAX | Stock No: 003771. RRID:IMSR_JAX:003771. | |
| Mouse line | *Olig2$^{Cre}$* | JAX | Stock No:025567. RRID:IMSR_JAX:025567. | |
| Mouse line | *P2ry12$^{CreER}$* | JAX | Stock No: 034727. RRID:IMSR_JAX:034727. | |
| Mouse line | *Pdgfrb$^{Cre}$* | JAX | MGI:6361145. | |
| Mouse line | *Pf4$^{Cre}$* | JAX | Stock No: 008535. RRID:IMSR_JAX:008535. | |
| Mouse line | *RG-Brainbow* | (Nakagawa et al. 2019) | Obtained from Dr. E. S. Anton. | |
| Mouse line | *Smad2 flox* | JAX | Stock No: 022074. RRID:IMSR_JAX:022074. | |
| Mouse line | *Smad3 flox* | JAX | MGI: 3822465. | |
| Mouse line | *Syn1$^{Cre}$* | JAX | Stock No: 003966. RRID:IMSR_JAX:003966. | |
| Mouse line | *Tgfb1 flox* | (Mo et al. 2007) | MGI:3719583. | |
| Mouse line | *Tgfb1 null* | (Kulkarni et al. 1993) | MGI: 1857671. | |
| Mouse line | *Tgfb1$^{GFP}$* | (Li et al. 2007) | MGI: 3719583. | |
| Mouse line | *Tgfbr2 flox* | (Chytil et al. 2002) | MGI: 2384513. | |
| Mouse line | *Tie2$^{Cre}$* | JAX | Stock No: 008863. RRID:IMSR_JAX:008863. | |
| Mouse line | *Ubc$^{CreER}$* | JAX | Stock No: 007001. RRID:IMSR_JAX:007001. | |
| Antibody | APOE | Abcam | Rabbit monoclonal. Cat# ab183596. RRID:AB_2832971. | Used at 1:300. |
| Antibody | CD206 | Biorad | Rat monoclonal antibody. Cat#: MCA2235T. RRID:AB_1101333. | Used at 1:150. |
| Antibody | CD11b-PE-Cy7 | eBioscience | Rat monoclonal antibody. Cat#: 50-154-54. RRID: AB_469588. | Used at 1:300. |
| Antibody | CD31 | R&D Systems | Goat polyclonal antibody. Cat# AF3628; RRID: AB_2161028. | Used at 1:300. |
| Antibody | CD45-APC | BD Biosciences | Rat monoclonal antibody. Catalog No: 559864. RRID:AB_398672. | Used at 1:100. |
| Antibody | CLEC7a | BioLegend | Rat monoclonal antibody. Cat#: 144302: RRID:AB_2561519. | Used at 1:300. |
| Antibody | FCRLS-APC | Butovsky lab | Rat monoclonal antibody. Clone 4G11. | Used at 1:1000. |
| Antibody | EGFP | Origene | Goat polyclonal antibody. Cat#: R1091P. RRID:AB_1002036. | Used at 1:2000. |
| Antibody | IB4-488 | Thermo Fisher | Cat#: I21411. RRID:AB_2314662. | Used at 1:150. |
| Antibody | IBA1 | Novus | Goat polyclonal antibody. Cat#:NB100-1028. RRID:AB_521594. | Used at 1:300. |
| Antibody | IBA1 | Wako | Rabbit polyclonal antibody. Cat#: 019-19741; RRID: AB_839504. | Used at 1:300. |
| Antibody | LGALS3 | CedarLane | Rat monoclonal antibody. Cat#:CL8942AP. RRID:AB_10060357. | Used at 1:150. |
| Antibody | Ly-6C-PerCP/Cy5.5 | Biolegend | Rat monoclonal antibody. Cat#: 128012. RRID:AB_1659241. | Used at 1:300. |
| Antibody | NESTIN-488 | Abcam | Mouse monoclonal antibody. Cat#: ab197495. RRID:AB_2732861. | Used at 1:150. |
| Antibody | NEUN | Millipore | Mouse monoclonal antibody. Cat#: MAB377. RRID:AB_2298772. | Used at 1:100. |
| Antibody | OLIG2 | Millipore | Rabbit polyclonal antibody. Cat#:: AB9610. RRID:AB_570666. | Used at 1:300. |
| Article | P2RY12 | Anaspec | Rabbit polyclonal antibody. Cat# 55043A. RRID:AB_2298886. | Used at 1:500. |

**Table 1 (continued) | Mouse Line and Reagent List**

| Reagent type | Designation | Source of references | Identifiers | Additional information |
|---|---|---|---|---|
| Antibody | P2RY12 | A generous gift from Dr. David Julius. | Rabbit polyclonal antibody. | Used at 1:1000. |
| Antibody | PDGFRA | R & D Systems | Goat polyclonal antibody. Cat#AF1062. RRID:AB_2236897. | Used at 1:300. |
| Antibody | GFAP | DAKO | Rabbit polyclonal antibody. Cat#Z0334.RRID:AB_10013382. | Used at 1:300. |
| Antibody | pSMAD3 (S423/S425) | Abcam | Rabbit monoclonal antibody. Cat# ab52903; RRID: AB_882596. | Used at 1:100. |
| Antibody | SOX9 | R and D Systems | Goat polyclonal antibody. Cat#: AF3075. RRID:AB_2194160. | Used at 1:300. |
| Antibody | TER119 | R&D Systems | Rat polyclonal antibody. Cat# MAB1125; RRID: AB_2297123. | Used at 1:150. |
| Antibody | TMEM119 | Abcam | Rabbit monoclonal antibody. Cat#: ab209064. RRID:AB_2800343. | Used at 1:100. |
| Molecular Biology Reagent | Opal secondaries | Okaya Biosciences | Cat#:FP1488001KT. | Used at 1:750 |
| Molecular Biology Reagent | RNAscope Multiplex Fluorescent Reagent Kit v2 | ACD Biosciences | Cat #323100. | |
| Molecular Biology Reagent | Percoll Plus | GE Healthcare | Cat#17-5445-02. | |

England Biolabs, E7775). Briefly, mRNAs were initially enriched with Oligod(T) beads. Enriched mRNAs were fragmented for 15 minutes at 94 °C. First strand and second strand cDNA were subsequently synthesized. cDNA fragments were end repaired and adenylated at 3'ends, and universal adapters were ligated to cDNA fragments, followed by index addition and library enrichment by PCR with limited cycles. The sequencing libraries were validated on the Agilent TapeStation, and quantified by using Qubit 2.0 Fluorometer as well as by quantitative PCR (KAPA Biosystems).

Ultra-low input RNA sequencing libraries (from sorted *Tgfb1$^{fl/fl}$; Cx3cr1$^{Cre}$* microglia) were prepared by using SMART-Seq HT kit for full-length cDNA synthesis and amplification (Takara, 634456), and Illumina Nextera XT (Illumina, FC-131-1096) library was used for sequencing library preparation. Briefly, cDNA was fragmented, and adapter was added using Transposase, followed by limited-cycle PCR to enrich and add index to the cDNA fragments. The sequencing library was validated on the Agilent TapeStation (Agilent Technologies, G2991BA), and quantified by using Qubit 2.0 Fluorometer (ThermoFisher, Q32866) as well as by quantitative PCR (KAPA Biosystems, KK4824). The sequencing libraries for bulk and ultra-low samples were multiplexed and clustered onto a flowcell on the Illumina NovaSeq instrument according to manufacturer's instructions. The samples were sequenced using a 2x150bp Paired End (PE) configuration. Image analysis and base calling were conducted by the NovaSeq Control Software (NCS). Raw sequence data (.bcl files) generated from Illumina NovaSeq was converted into fastq files and de-multiplexed using Illumina bcl2fastq 2.20 software. One mis-match was allowed for index sequence identification.

## Imaging

Confocal images were taken with 10x, 20x or 40x objectives using a motorized Zeiss 780 upright laser scanning confocal microscope or a motorize Zeiss 900 inverted laser scanning confocal microscope. Image brightness and contrast was optimized in ImageJ. Images and figures were arranged in Adobe Illustrator.

## Immunofluorescence quantification

Confocal images were taken of immunostained brains as described above. For CD31/TER119 stains, single images were taken, whereas z-stacks of three optical sections were taken for P2RY12/IBA1 stains. Each genetic condition was at least n = 3, and three images were taken

of each mouse and/or brain region being studied (e.g., cortex vs. striatum). Z stacks with at least 3 images per stack were converted into maximum intensity projections using Fiji/ImageJ. Cells that were positive for a given marker were manually counted using the cell counter plugin of Fiji.

## Statistics and reproducibility

Sample size was not precalculated using a statistical framework, but conformed to general standards in the field. Sample analysis was not blinded. We generally performed all experiments on both sexes with equal numbers of age-matched mice (number n, indicated in figure legends), with a minimum of 3 samples for imaging-based observations, a minimum of 4 samples for RNA analysis and a minimum of 5 for behavioral experiments. Sample sizes were based on our previous work[16,78,79] and provide sufficient statistical power to achieve precise and accurate conclusions. Data analysis was tested for normal non-normal distribution. We used two-tailed Student's t-test with 95% confidence for comparing two conditions, two-way ANOVA for multiple comparisons, and independent cohorts for reproducibility. For all immunostaining quantification, values for each mouse were calculated by averaging three pictures from each mouse. Differences between means were compared using a two-tailed t-test, with an alpha of 0.05.

## ATAC-seq

ATAC-seq was carried out following the Omni-ATAC-seq protocol[80]. Briefly, 50k cortical microglia (CD11b$^+$Fcrls$^+$ Ly-6C$^−$) from seven *Itgb8$^{fl/fl}$;Emx1$^{Cre}$* P60-P90 mouse brains and seven control littermates were sorted into Eppendorf tubes by a BD FACSAria™ II (BD Bioscience). Samples from each mouse served as a separate biological and experimental replicate. After being washed with cold dPBS, microglia were permeabilized with ATAC-Resuspension Buffer containing 0.1% NP40, 0.1% Tween-20, and 0.01% Digitonin and washed with ATAC-RSB buffer with 0.1% Tween-20. The isolated microglial nuclei were then incubated in transposition mixture at 37 °C for 60 min. The DNA were purified using a Zymo DNA clean and concentrator-5 kit (Zymo research, D4014) and amplified using NEBNext® High-Fidelity 2x PCR Master Mix (NEB, M0541s) for 9 cycles. The ATAC libraries were size selected using SPRIselect beads (Beckman Colter, B23317) and quantified using Agilent 2100 Bioanalyzer and Qubit (Invitrogen, Q33238). The pooled libraries were

then sent to Azenta Life Sciences for paired-end sequencing (2×150 bp) on Illumina HiSeq platform.

## ChIP-seq
Anti H3K9ac ChIP-seq were performed using using the iDeal ChIP-seq kit for Histones (Diagenode, C01010059) and NEBNext® UltraTM II DNA Library Prep Kit for Illumina® (NEB, E7103S). Briefly, microglia were sorted from *Itgb8*<sup>fl/fl</sup>*;Emx1*<sup>Cre</sup> mouse brain cortex and were pooled from 3-4 different individuals with same genotype to achieve >200 k cells per biological replicate. 3 mutant and 4 littermate control (*Emx1*<sup>Cre(−)</sup>*;Itgb8*<sup>fl/fl</sup> samples were generated in this way, from a total of 35 mice. The microglia were fixed with 1% formaldehyde at 20 °C for 8 min then quenched with 0.125 M Glycine. The nuclei were isolated and sonicated using a Diagenode disruptor for 20 cycles (30 seconds ON, 30 seconds OFF, high power). The chromatin fragments were enriched using 2ul of anti-H3K9ac antibody (Millipore, 07-352) per reaction. The ChIPed DNA was purified and processed for library preparation. The ChIP DNA libraries were size selected using SPRIselect beads (Beckman Colter, B23317) and quantified using Agilent 2100 Bioanalyzer and Qubit (Invitrogen, Q33238). The pooled libraries were then sent to Azenta Life Sciences for paired-end sequencing (2×150 bp) on Illumina HiSeq platform.

## Epigenetic data analysis
Raw fastq files were first checked for quality using Multiqc sequence analysis. Cutadapt (v.4.0) was used to cut adapters (-*a* AGATCGGAA-GAGCACACGTCTGAACTCCAGTC -*A* AGATCGGAAGAGCGTCGTGTAG GGAAAGAGTGT), and reads were aligned to the mouse genome (mm10) using bowtie2 (2.3.4.3)[81]. Subsequently, SAM files were sorted and filtered using Sambamba (v.0.8.2)[82]. We discarded unmapped and duplicate fragments. Sorted BAM files were indexed using samtools (v.1.15.1)[83]. Normalization for all BigWig files were carried out against effective genome size (2652783500 for mice H3K9ac ChIP-seq and ATAC-seq). Peak analysis was carried out with Macs2 (v.2.2.7)[84] using stringent parameters (-f BAMPE --mfold 5 50 -p 0.001). A master-peak file was created, using the following settings for ATAC (-d 300 -c 7) and H3K9ac (d 300 -c 3). Raw counts were extracted from BAM files using master-peak file and loaded into DESeq2[85]. Differentially accessible peaks were determined with adjustment for false discovery using benjamini hochberg method (Padj < 0.05). Differentially accessible peaks were annotated using HOMER (v4.4)[45]. BAM files were then converted to BigWig files and merged per experimental group for visualization using deeptools (v.3.5.0)[86]. Peak plots were constructed using deeptools with a 1000 bp extension from the center of the peak. Peaks were visualized using IGV (v.2.11.4), exported as.png, and edited in Adobe Illustrator.

## Public data comparison
ATAC-seq raw data from GSE79816[43] were downloaded from GEO and analyzed according to methods above, apart from peak-calling. Peak information was used from *Itgb8*<sup>fl/fl</sup>*;Emx1*<sup>Cre</sup> experiment for comparison. Differential peaks were determined comparing each microglia ATAC-seq timepoint (Yolk-sac, E12.5 and Pre-Microglia) to the adult microglia ATAC-seq profile. Differential peaks were then compared against differential peaks between *Itgb8*<sup>fl/fl</sup>*;Emx1*<sup>Cre</sup> microglia and WT microglia.

## Reporting summary
Further information on research design is available in the Nature Portfolio Reporting Summary linked to this article.

## Data availability
Source data are provided with this paper. The sequencing data generated in this study have been deposited in the GEO database under accession codes as follows: ChIP-seq data have been deposited with the accession code GSE242221 (RNA-seq data (whole brain data, isolated *Tgfb1* mutant microglia datasets) have been deposited with the accession GSE236615. Public ATAC-seq data was obtained from GSE79816. *Itgb8*<sup>fl/fl</sup>*;Emx1*<sup>Cre</sup> microglia and *Smad2/3*<sup>fl/fl</sup>*;Cx3cr1*<sup>Cre</sup> data were generated previously by our group and are available under GEO GSE239603. *Tgfbr2*<sup>fl/fl</sup>*;Cx3cr1*<sup>Cre</sup> microglia is from GEO GSE124868. Data from *Lrrc33*<sup>−/−</sup> microglia and whole brain was obtained from GEO GSE96938. Source data are provided with this paper.

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

## Acknowledgements

This study was in supported in part by HDFCC Laboratory for Cell Analysis Shared Resource Facility through a grant from NIH (P30CA82103). This research was supported, in part, by NIH grants R01NS123168 (T.D.A.) and the UCSF Vision Core shared resource of the NIH/NEI P30 EY002162. H.P. was supported in part by the French Agence Nationale de la Recherche (ANR) under grant ANR-20-CE15-0015. X.Z. was supported by the National Natural Science Foundation of China (82201581). Support for cartoon drawing was provided by Biorender https://BioRender.com/r69o433. We would like to thank Dr. Richard Flavell for providing *Tgfb1^flox* and *Tgfb1^GFP* mice. We would also like to thank Dr. Rosemary Ackhurst for providing *Tgfb1* null mice.

## Author contributions

T.A. supervised the study, performed the behavioral analysis in Figure S1 and did the cell counting in Figure S1. G.L.M. designed the study, under the supervision of T.A. G.L.M. performed the histological analyzes, with specific contribution from others listed here. N.S. performed preliminary histological analysis of postnatal *Tgfb1^fl/fl^/Tgfb1^F/GFP^;Cx3cr1^Cre* and *Smad2/3^fl/fl^;Cx3cr1^Cre* mice. G.L.M performed the imaging, with support from T.A. for Figure S2. Mouse husbandry was done by G.L.M., with support from N.S., who initially bred *Tgfb1^fl/fl^/Tgfb1^F/GFP^;Cx3cr1^Cre* and *Smad2/3^fl/fl^;Cx3cr1^Cre* mice, and with support from A.K., L.T., M.B., K.C, H.L., K.W., J.H.K. and D.M. D.M. also performed Western blot analyzes. G.L.M. performed RNA isolation and bulk RNA-seq experiments for Fig. 7. N.S. performed bioinformatic analysis in Figs. 2, 7, Supplementary Figs. 6,7 and 8. X.Z. performed RNA-seq, ATAC-seq and ChIP-seq analysis of sorted microglia in *Itgb8^fl/fl^;Emx1^Cre* mice for Figs. 2 and 3, and Supplemental Fig. 5, with bioinformatic support from K.K. and with conceptual input from O.B. A.K and L.T sectioned *Tie2^Cre^* and *Pdgfrb^Cre^* embryos used in Fig. 3. G.L.M. performed the flow cytometry experiments to isolate microglia from *Tgfb1^fl/fl^; Cx3cr1^Cre* mice, with support from C.O. M.L., created the graphical abstract and the Fig. 6 illustration, contributed to image processing, cell quantification and helped to edit and revise the manuscript. C.O. performed the flow cytometry experiments to isolate microglia from *Smad2/3^fl/fl^; Cx3cr1^Cre* mice and performed CD31/TER119 quantification, with support from A.K. L.T. collected and stained embryos used in the *Emx1^Cre^; RG-Brainbow* analysis. J.H.K. helped to collect *Nestin^Cre^* embryos used in Fig. 1. H.P. and E.S. provided experimental support with the *Itgb8^tdT^* and *RG-Brainbow* mouse lines, respectively. D.S. provided input regarding experimental design and interpretation. O.B. supervised work by X.Z. and K.K. Figures were generated by G.L.M., with input from T.A. The manuscript was written by G.L.M and T.A.

## Competing interests

Drs Butovsky, Arnold and Sheppard have a financial interest in Glial Therapeutics, a company developing a new therapy to target ITGB8-TGFb signaling as a treatment for Alzheimer's disease. Dr. Butovsky's interests were reviewed and are managed by BWH and Mass General Brigham in accordance with their conflict of interest policies. The remaining authors declare no competing interests.
