## [Transparent Peer Review file · Nature Communications]

Neural stem cell integrin $\alpha\text{V}\beta\text{8}$ regulates cell autonomous microglial TGF β 1 signaling that is necessary for microglial identity

Corresponding Author: Dr Gabriel McKinsey

This manuscript has been previously reviewed at another journal that is not operating a transparent peer review scheme. The manuscript was considered suitable for publication without further review at Nature Communications.

Version 0:

Reviewer comments:

Reviewer #1

(Remarks to the Author)

This is a very well executed and elegant study that addressed the role of autocrine TGF β 1 signaling in microglia development. I have no major comments and I recommend publication. Congratulations to the authors.

Reviewer #2

(Remarks to the Author)

Here, McKinsey et al. extend the scholarly studies on microglial development and homeostasis focusing on integrin $\alpha\text{V}\beta\text{8}$ -TGF β 1 signaling that they have been conducting for several years, providing solid data that helps elucidate the cellular and molecular mechanism. The authors utilize a series of conditional knockouts to propose a model where $\alpha\text{V}\beta\text{8}$ expression on radial glia activates latent TGF β 1 on microglial precursors. Autocrine TGF β 1 signaling then promotes microglial maturation. The domain-restricted microglial defects induced by embryonic deletion of $\alpha\text{V}\beta\text{8}$ in radial glia are an interesting new model and represent a novel approach. The research highlights developmental pathways that shape long-term microglial identity, which may have eventual implications for neurodevelopmental disorders. Although the present findings incrementally advance our understanding of previously established principles regarding microglia development, the overall concept of $\alpha\text{V}\beta\text{8}$ -TGF β 1 regulation of microglia is not entirely novel, as prior work has implicated this pathway in microglia. Thus, the major point of the study is the identification of how, where and when $\alpha\text{V}\beta\text{8}$ -TGF β 1 pathway is activated to make microglia get differentiated and matured during development. The conditional knockout and histological methods are appropriate, but some supporting datasets lack statistical rigor. Additionally, further insight is needed into the proposed non-canonical TGF β signaling effects on microglia. The major findings appear reproducible, contingent on additional replicates and statistical testing to ensure robustness.

In summary, this study presents some interesting new concepts but lacks mechanistic depth in areas. Addressing the major concerns would strengthen the manuscript.

Major points and suggestions

1. The weakness is that the involvement of early radial glia (E10.5-E13.5) in microglia differentiation is shown only based on circumstantial evidence, although it is plausible. I do understand the technical limitation, but at least the authors should confirm the efficient deletion of *Itgb8* in radial glia at this phase.
2. The authors show that $\alpha\text{V}\beta\text{8}$ expression on radial glia activates latent TGF β 1 on microglial precursors, which contributes to microglia differentiation in cortex and hippocampus. What about the other brain areas? Could the same mechanism be involved in microglia maturation? Answering this is important to boost the broad interests of the paper.
3. Figure 1G,H shows analysis of *Emx1-Cre*;RG-Brainbow mice suggesting contact between radial glia endfeet and CD206+ macrophages in the meninges. To conclusively demonstrate that CD206+ precursor cells interact with radial glia, resulting in transition into CD206- microglia in the parenchyma, intravital imaging is essential.
4. The paper would benefit from additional quantitative analysis of the *Itgb8* deletion effects in Figures 1D-F. Similarly, inclusion of control quantifications in Figures 4C-G and 5 is recommended to statistically demonstrate the impact of *Itgb8*

loss.

For Figure 2C-G, adding quantitative data on marker expression in control vs *Itgb8* knockout mice would enhance the evidence.

Minor comments:

1. Please carefully proofread figure labels, legends, and terminology for consistency. For example, Figure 1G shows "Nestin-cre" but the legend and text refer to *Emx1-cre*, suggesting the Fig 1G label is incorrect.
2. Consider adding Ai14 reporter data to Figures 4E and 4G as in 4D.
3. The reference to "Figure 1D" under the section "*Tgfb1* is required cell-autonomously for microglial development" appears incorrect, as it should be Figure 4D.
4. Please unify RNAseq/RNASeq terminology as RNA-seq throughout.
5. In Figure 3B, the gene names are very small and overlapping, making it difficult to discern which dots represent which genes. Enlarging the text would improve clarity.
6. The #1 and #2 regions indicated in Figures 6B-G could be clarified by noting the specific brain regions in the legend or directly labeling them on the figure panels.

Reviewer #3

(Remarks to the Author)

This is a review for the manuscript title "Integrin α V β 8-TGF β 1 signaling in microglial development and homeostasis" submitted to the journal Nature Communications

In this manuscript the authors present the results of a series of studies aiming to identify 5 aims: 1) the specific cell types in the mouse brain that are relevant for microglial TGF β signaling, 2) the timing of α V β 8-TGF β 1 interactions, 3) the source of microglial TGF β , 4) the relationship between dysmature microglia and DAM's, and 5) the role of canonical versus non-canonical TGF β signaling. To answer these questions the authors utilized a wide array of genetic mouse lines and a combination of Immunohistochemistry, Flow cytometry, RNA sequencing, ATAC-seq, and ChIP-seq to phenotype each genetic model with the goal of characterizing TGF β signaling in microglia. The authors conclude that microglial differentiation is dependent upon the activation of TGF β 1 by *Itgb8*-expressing radial glia progenitors, identify embryonic days 9-13.5 as critical, and that signaling largely depends on the Smad pathway. The authors then speculate about the usefulness of these findings in creating new models of region restricted microglial dysfunction and in studying disease states. While the studies described in this manuscript represent a significant amount of work, the manuscript requires substantial revision to include all relevant controls and both animal and statistical information before a conclusion can be drawn about the results of the studies. Overall, the manuscript lacks a coherent overall story and is often difficult to follow. The lack of wildtype littermate controls and image quantification makes it impossible to accurately assess the stated claims. The paper also fails to include sex as a biological variable in any experiment, even though this is an incredibly important aspect of microglia development and function.

Major comments:

1. There is a lack of image quantification across the entire manuscript that makes the results difficult to interpret. For example, in Figure 1 there is no validation of the *Itgb8* knockout in the specific target cell populations, if this is consistent across the brain or even where the images were taken from. There are no wildtype littermate controls. Figures 4, 5 6 and 7 also require quantification for all stains. For example, in figure 4, what is the percentage of deletion in the different *Tgfb1* flox line manipulations? The vast majority of the imaging based figures are similar in their lack of proper controls, quantitative data and statistics.

2. The figures often do not substantiate the claims made in the figure legend or text. For example, Figure 1 does not demonstrate that microglial maturation is dependent on interactions with radial glia. There are two example images, no quantification, no statistics and no genotype differences shown. How common are the interactions between embryonic radial glia and microglia precursors (Figure 1H)? Quantitative data demonstrating that these contacts happen more often than expected by chance would be required to make this claim "In total, these data suggest that *Itgb8* expression in early embryonic radial glia is necessary for the TGF β -dependent differentiation of CD206+ microglial precursors, and that in the absence of *Itgb8*, these CD206+ precursors populate the brain, but fail to express microglial-specific markers." It would also be necessary to actually look at additional timepoints to demonstrate that the CD206+ precursors do populate the brain as expected but do not show microglial markers. Is it possible that these precursors are actually BAMs?

As another example, in the discussion the claim is made that "Our analysis shows that this is reflected by widespread epigenetic differences in homeostatic, MGnD/DAM and BAM gene loci. These epigenetic differences are long-lasting and stable through adulthood, beyond the point at which *Itgb8* appears to be necessary for microglial homeostasis." This isn't shown in the paper as there was only an adult timepoint done for analysis and it is not clear when the epigenetic differences arise. These types of unsubstantiated claims occur throughout the manuscript.

3. The presentation of the transcriptomics and epigenomics datasets are not clear. For example, is the heatmap in Figure 2 scaled in some way? The legend lacks labels and it is unclear how the statistics were done to assess significant enrichment throughout this figure. Are these gene expression changes unique to developmentally regulated genes or are other genes

also misregulated? It is unclear if this data was generated for this paper or retrieved from GEO from another publication. If this is the first analysis of the dataset, a standard differential gene expression analysis would be the appropriate first step in the analysis. There is no statistical testing of how the control and *Itgb8* microglia change in the target gene expression in Figure 2H. Statistics need to be clearly defined that include corrections for multiple comparisons and significant p-values for individual genes reported.

Similarly, for Figure 3 it is unclear how widespread the chromatin accessibility and H3K9ac differences are in the wildtype compared to *Itgb8^{fl/fl};Emx1Cre* dysmature microglia. Having a standard volcano plot with significant regions depicted would be required to visualize the extent of the chromatin and acetylation changes. It is also difficult to access the magnitude of change in the heatmap in 3D and 3E as they appear scaled in some way that is not clearly defined in the legend. The correlations in Figure 3B are also hard to read as many of the gene names overlap. Please also include all differential gene, region and acetylation lists in a supplemental excel file to make it easy for readers to examine genes and regions of interest. A UCSC browser track hub or bigwig files on GEO would also help for better data access for readers.

Similarly, for Figure 7 it is unclear how Figure 7C was constructed. Are the correlations done in terms of log fold change in expression differences between wildtype and mutant for each condition? Or some other metric? It would be nice to see how many differentially expressed genes are shared or unique among the mutants – potentially using something like an upset plot. Figure 7E lacks a legend and statistics. Do any of the genes represented reach statistical threshold for significance after FDR correction? The overlaps with ATM and PAM as mentioned in the text are not shown.

4. The writing is very field specific, and jargon filled making it difficult to read and unclear what the overall significance of the work is for readers in a broader audience. In the introduction, a first paragraph on the large importance of this area of work would be helpful for framing the study for the diverse readership of Nature communications. The introduction is also incredibly dense and very full of jargon. For example, it isn't clear that *avB8* is the same as *Itgb8*. The figure diagram in figure 7A helps clarify this but doesn't come until the end of the paper. Moving the panel from 7A to figure 1 and adding in the canonical vs non-canonical signal details would help. There are also lots of different transgenic manipulations and staining conditions across these figures and better labeling and/or diagrams would help readers follow the logic of the conditions more easily. The graphical abstract also implies that the work is heavily focused on microglia-radial glia interactions in development which is misleading given this is only one panel of figure 1 and there isn't any quantitative data on this point.

5. Sex is not included as a biological variable in any of the experiments. Sex has an important influence on microglial development (see PMID: 31743527 for a review) and is a critical mediator of disease risk. The sex of the animals is not stated for any of the experiments and all experiments should have both male and female animals included and analyzed for sex differences.

Minor comments:

1. In multiple figures, the dotted blow-up box doesn't match the small box. Everything needs scale bars and pink boxes over red staining is not easy to see. There are also multiple problems with the labeling of figures and legends.

2. Figure 2A. A white, aged mouse is shown in the figure, but black mice were used throughout the study. Suggest replacing for clarity.

3. The figure legend for Figure 2 appears to be wrong, the title doesn't match the text and the individual panels are not described sufficiently.

4. In Figure 4A, what causes the vascular defects and hemorrhage? weaker expression in brain blood vessels (both endothelial cells and PDGFR β + mural cells) at E14.5.

5. In Figure 6, is the patchiness of *Tgfb1* loss of expression due to failure in recombination? Please quantify the number of microglia with recombination and compare phenotypes of recombined and non-recombined microglia quantitatively for the different stains.

6. Figure 7F requires quantification of LGALS3 levels, preferably by more quantitative methods such as western blot on sorted microglia or intracellular flow cytometry.

7. Neuromotor phenotypes are missing from Figure S8 and the immune staining figures lack quantification.

8. The methods lack information on mouse husbandry including light cycle, diet etc. Control genotypes are not clearly stated for each experiment nor is information on breeding strategy.

9. The fluorescence-activated cell sorting is based on several microglial markers (two are explicitly listed in the methods, others are unclear). Are these markers significantly altered in the various mutants? Please include the sort data and yields to assess purity and potential impacts on microglial survival.

Reviewer #4

(Remarks to the Author)

I co-reviewed this manuscript with one of the reviewers who provided the listed reports as part of the Nature Communications initiative to facilitate training in peer review and appropriate recognition for co-reviewers.

Version 1:

Reviewer comments:

Reviewer #2

(Remarks to the Author)

The authors addressed most of the reviewer's concerns and the modifications made including the new data have strengthened the manuscript and its conclusions. Nevertheless, some points need to be further addressed.

1. For Figure 1D-H, the authors should provide images of the control group that correspond to the quantitative data presented.
2. There appears to be an inconsistency between the quantitative data presented in Figure 2G and the corresponding figures 2E and F. It seems that the data for the cortex and striatum may have been reversed.
3. There is a discrepancy in the numbering of the supplementary figures. Please verify and correct the numbering of the figure starting with "Gene Ontology and..."
4. Please ensure that all references in the main text are updated accordingly.

Reviewer #3

(Remarks to the Author)

Summary:

The manuscript is much improved from the previous submission but still requires several major and minor revisions before publication.

Major Concerns:

1. For lines 293-317, this experiment uses P30 mice, which are adolescent not adult. The manipulations were also done on P4-6 which is developmental, not adult. Claims about adult homeostasis would require manipulations in the adult mouse (P80 or older).
2. Several figures still lack image quantification. For example, Figure 6 requires quantification. Figure S10 is missing the neuromotor data described in line 373 and the images require quantification to make the claim that there are now major impacts on microglial phenotypes (line 373). Figure 7F-H requires quantification. The whole brain western (S11) is not sufficient quantification to draw conclusions about changes in the microglia.
3. Sex is still not adequately addressed as a biological variable. Only three biological replicates are shown for each quantification, which is not sufficient to address sex differences. Please clarify which sex each data point is in each figure and discuss this as a limitation in the discussion.

Minor Concerns:

Please revise the working in Supplemental figure 2. Brain-wide deletion of *Itgb8* with NestinCre results in loss microglial of microglial homeostasis and increased reactivity, as shown by loss of P2RY12 expression and upregulation of the reactive marker LGALS3.

Supplemental figure 4L, the open arrow in the #2 blow up is not obviously pointing at an IBA1+ cell.

To strengthen the claim that the dysmature microglia are similar to immature microglia, it would help to compare the ATAC-seq signatures to one of several developmental profiling studies with ATAC-seq across early microglial development such as (Matcovitch-Natan et al., 2016; Thion et al., 2018). It is less clear that the epigenetic changes are "associated with MGnD and BAM gene upregulation" as stated in the title of this section (line 210-211). Please either revise the language and remove the adult part or provide evidence in adults.

Figure 4C and D, please check the arrow placements on the blow-outs. They should be consistent across panels, but they are not.

Line 288 references 5G. please double check that this is correct.

Line 297: "(how much intragastric)" should be an actual amount given

Line 302. What are "b1-intact microglia"?

There are two supplemental 7 labelled figures.

Reviewer #4

(Remarks to the Author)

Reviewer #1 (Remarks to the Author):

This is a very well executed and elegant study that addressed the role of autocrine TGF β 1 signaling in microglia development. I have no major comments and I recommend publication. Congratulations to the authors.

We very much appreciate the reviewer's positive appraisal of our work!

Reviewer #2 (Remarks to the Author):

Here, McKinsey et al. extend the scholarly studies on microglial development and homeostasis focusing on integrin α V β 8-TGF β 1 signaling that they have been conducting for several years, providing solid data that helps elucidate the cellular and molecular mechanism. The authors utilize a series of conditional knockouts to propose a model where α V β 8 expression on radial glia activates latent TGF β 1 on microglial precursors. Autocrine TGF β 1 signaling then promotes microglial maturation. The domain-restricted microglial defects induced by embryonic deletion of α V β 8 in radial glia are an interesting new model and represent a novel approach. The research highlights developmental pathways that shape long-term microglial identity, which may have eventual implications for neurodevelopmental disorders. Although the present findings incrementally advance our understanding of previously established principles regarding microglia development, the overall concept of α V β 8-TGF β 1 regulation of microglia is not entirely novel, as prior work has implicated this pathway in microglia. Thus, the major point of the study is the identification of how, where and when α V β 8-TGF β 1 pathway is activated to make microglia get differentiated and matured during development. The conditional knockout and histological methods are appropriate, but some supporting datasets lack statistical rigor. Additionally, further insight is needed into the proposed non-canonical TGF β signaling effects on microglia. The major findings appear reproducible, contingent on additional replicates and statistical testing to ensure robustness.

In summary, this study presents some interesting new concepts but lacks mechanistic depth in areas. Addressing the major concerns would strengthen the manuscript.

We are grateful for the reviewer's positive evaluation of our work and constructive criticisms. We agree with the suggestion that adding additional statistical rigor would greatly strengthen the paper, and we have focused a great deal of our subsequent work on adding this to the manuscript.

In terms of general comment on "lack of statistical rigor": we generally performed all experiments on both sexes with equal numbers of age-matched mice (number n, indicated in figure legends), with a minimum of 3 samples for imaging-based observations, a minimum of 4 samples for RNA analysis and a minimum of 5 for behavioral experiments. Sample sizes were based on our previous work¹⁻³ and provide sufficient statistical power to achieve precise and accurate conclusions. Data analysis was tested for normal non-normal distribution. We used two-tailed Student's t-test with 95% confidence for comparing two conditions, two-way ANOVA for multiple comparisons, and independent cohorts for reproducibility. We have added this information on our study's rigor in our methods section.

Regarding the reviewer's general comment on the need for "further insight into proposed non-canonical TGF β signaling effects on microglia," we have added references and additional discussion of this topic in our Discussion section.

We have also made the following revisions to the manuscript, addressing all of the specific comments below.

Major points and suggestions:

*1. The weakness is that the involvement of early radial glia (E10.5-E13.5) in microglia differentiation is shown only based on circumstantial evidence, although it is plausible. I do understand the technical limitation, but at least the authors should confirm the efficient deletion of *Itgb8* in radial glia at this phase.*

We thank the reviewer for this suggestion. Indeed, it is challenging to unequivocally demonstrate involvement of early radial glia versus neuroepithelial cells or downstream progeny, and we appreciate the recommendation to validate knockout in lieu of more direct evidence. To that end, we have performed western blot analyses from brain cortical lysates in *Itgb8^{fl/fl};Emx1^{Cre}*, *Itgb8^{fl/fl};hGFAP^{Cre}* mutants and *Itgb8^{fl/+};Emx1^{Cre}* controls. This analysis is included in an updated Supplementary Figure 4. These results reveal that in the E14.5 cerebral cortex, *Itgb8^{fl/fl};Emx1^{Cre}* show very little residual ITGB8 protein (85% reduced), while we found that there is a significant ($p = 0.008$) and substantial (58%) reduction in the cortex of *Itgb8^{fl/fl};hGFAP^{Cre}* mutants. In total, our systematic recombination analysis with many different lineage restricted Cre and CreER lines (Supplemental Figure 1) suggest that *Itgb8* expression in postmitotic cell lineages is dispensable for microglial differentiation. Conversely, our embryonic and postnatal analysis of *Itgb8* deletion with *Nestin^{Cre}*, *Emx^{Cre}*, *hGFAP^{Cre}* lines (Figure 1, Supplemental Figure 1), suggest that *Itgb8* in early neural progenitors is required for microglial differentiation.

2. The authors show that α V β 8 expression on radial glia activates latent TGF β 1 on microglial precursors, which contributes to microglia differentiation in cortex and hippocampus. What about the other brain areas? Could the same mechanism be involved in microglia maturation? Answering this is important to boost the broad interests of the paper.

We are grateful to the Reviewer for raising these interesting questions and helping to shape the paper for a broader audience. In response, we have analyzed other areas of the central nervous system of *Itgb8^{fl/fl};Nestin^{Cre}* mice in adult mice (please see new Supplementary Figure 2) for evidence of a microglial phenotype similar to that observed in the cortex and hippocampus. We chose *Nestin^{Cre}* for its early and broad recombination across many regions of the CNS. As with our analysis in the cortex and the hippocampus, we find that dysmature microglia are present throughout the brain (Supplementary Figure 2B) and spine (Supplementary Figure 2D). Together, these data support our model that *Itgb8* is involved in microglial development throughout the nervous system.

3. Figure 1G,H shows analysis of Emx1-Cre;RG-Brainbow mice suggesting contact between radial glia endfeet and CD206+ macrophages in the meninges. To conclusively demonstrate that CD206+ precursor cells interact with radial glia, resulting in transition into CD206- microglia in the parenchyma, intravital imaging is essential.

We agree that intravital imaging would offer elegant orthogonal support of our model, and would like to pursue this for future experiments. However, at the moment this technique is not readily available to our laboratory and we believe that further study in this regard is a future direction, beyond the scope of the current manuscript.

The origin of microglia from CD206⁺ progenitors is a particularly important and novel concept for the field. We previously hypothesized this based on anatomical and histological observations (please see ⁴). Recently, multiple groups have conclusively shown that CD206⁺ macrophages are the precursors to CD206⁻;P2RY12⁺ microglia (Please see ⁵ and ⁶). This was done via genetic fate mapping from CD206⁺ embryonic macrophages, using *CD206^{CreER}* recombination as a method to label and fate-map CD206⁺ embryonic macrophages. We believe that our findings regarding the role of *Itgb8* in microglial development are complementary to these findings. Specifically, our findings suggest a cell-cell signaling mechanism that is necessary for the transformation of a CD206⁺ precursor into a CD206⁻;P2RY12⁺ microglia and that in the absence of this signaling, CD206⁺ microglial precursors are trapped in an immature state, although they do still populate the brain.

4. The paper would benefit from additional quantitative analysis of the Itgb8 deletion effects in Figures 1D-F. Similarly, inclusion of control quantifications in Figures 4C-G and 5 is recommended to statistically demonstrate the impact of Itgb8 (and Tgfb1) loss. For Figure 2C-G, adding quantitative data on marker expression in control vs Itgb8 knockout mice would enhance the evidence.

We thank the reviewer for their suggestion regarding quantification and statistical analysis. This was also requested by Reviewer #3. We have performed extensive quantitative and statistical analyses throughout the

manuscript to support main figures 1,2,4,5, and 7, as well as Supplementary Figures 4,5,8,9,10. We agree that the addition of these data greatly bolsters the strength of our model and lends additional scientific rigor to the paper. Thank you for the suggestion.

For Figures 1, 4, and 5, we quantified extra-luminal TER119+ erythrocytes and loss of P2RY12 staining to reflect vascular/hemorrhagic and microglial phenotypes respectively. For all of these analyses, the quantification strongly supports our qualitative findings as shown by representative immunohistochemistry images.

For Figure 2, we added quantification of striatal versus cortical microglial phenotype in *Emx1^{Cre}* mutants, highlighting microglial changes in the affected mutant cortex, but not in the Cre- non-recombined control striatal region of the brain.

To support the conclusions of Figure 7, in Supplementary Figure 11, we added western blot quantification of LGALS3 expression in *Cx3cr1^{Cre};Tgfb^{fl/+}* controls, *Cx3cr1^{Cre};Tgfb^{fl/fl}* mutants, and *Cx3cr1^{Cre}; Smad2/3^{fl/fl}* mutants to quantitatively assess our immunofluorescence findings.

In addition to the main text figures, we have also added quantification and new data visualizations to Supplementary Figures 4,5,8,9 and 10. To summarize:

- Supplementary Figure 4, contains quantification of ITGB8 protein expression changes in conditional *Itgb8* mutant models as measured by western blot.
- Supplementary Figure 5 contains a newly generated volcano plot display of ATAC-seq and H3K9ac ChIP-seq data, in response to a request from Reviewer 3.
- In response to a request from Reviewer 3, we have added an upset plot of gene expression data to Supplementary Figure 8.
- Supplementary Figure 9 includes an analysis of sex-specific phenotypes in an *Lrrc33* knockout model, in response to a request from Reviewer #3.

Minor comments:

1. Please carefully proofread figure labels, legends, and terminology for consistency. For example, Figure 1G shows "Nestin-cre" but the legend and text refer to *Emx1-cre*, suggesting the Fig 1G label is incorrect.

We apologize for this oversight and thank the reviewer for catching this error. We have now systematically examined the manuscript to proofread figure labels, legends and terminology for consistency, and corrected any errors, including the error in Fig 1G.

2. Consider adding *Ai14* reporter data to Figures 4E and 4G as in 4D.

We apologize for any confusion. Figures 4F and 4H are *Tgfb1^(fl-);P2ry12^{CreER}* and *Tgfb1^(fl-);Pf4^{Cre}* mutant mice, respectively, and both contain the *Ai14* tdT reporter. We have added the *Ai14* nomenclature to row labels.

3. The reference to "Figure 1D" under the section "Tgfb1 is required cell-autonomously for microglial development" appears incorrect, as it should be Figure 4D.

We apologize for this mistake and have fixed this error.

4. Please unify RNAseq/RNASeq terminology as RNA-seq throughout.

We have unified our terminology and apologize for the inconsistency.

5. In Figure 3B, the gene names are very small and overlapping, making it difficult to discern which dots represent which genes. Enlarging the text would improve clarity.

We thank the reviewer for this observation. We agree that the text was too small and jumbled. We have reduced the number of listed genes and have increased the text size of remaining gene names.

6. The #1 and #2 regions indicated in Figures 6B-G could be clarified by noting the specific brain regions in the legend or directly labeling them on the figure panels.

These images were taken from the cerebral cortex. We have updated Figure 6 Legend to note this.

Reviewer #3 (Remarks to the Author):

This is a review for the manuscript title “Integrin α V β 8-TGF β 1 signaling in microglial development and homeostasis” submitted to the journal Nature Communications. In this manuscript the authors present the results of a series of studies aiming to identify 5 aims: 1) the specific cell types in the mouse brain that are relevant for microglial TGF β signaling, 2) the timing of α V β 8-TGF β 1 interactions, 3) the source of microglial TGF β , 4) the relationship between dysmature microglia and DAM’s, and 5) the role of canonical versus non-canonical TGF β signaling. To answer these questions the authors utilized a wide array of genetic mouse lines and a combination of Immunohistochemistry, Flow cytometry, RNA sequencing, ATAC-seq, and ChIP-seq to phenotype each genetic model with the goal of characterizing TGF β signaling in microglia. The authors conclude that microglial differentiation is dependent upon the activation of TGF β 1 by Itgb8-expressing radial glia progenitors, identify embryonic days 9-13.5 as critical, and that signaling largely depends on the Smad pathway. The authors then speculate about the usefulness of these findings in creating new models of region restricted microglial dysfunction and in studying disease states. While the studies described in this manuscript represent a significant amount of work, the manuscript requires substantial revision to include all relevant controls and both animal and statistical information before a conclusion can be drawn about the results of the studies. Overall, the manuscript lacks a coherent overall story and is often difficult to follow. The lack of wildtype littermate controls and image quantification makes it impossible to accurately assess the stated claims. The paper also fails to include sex as a biological variable in any experiment, even though this is an incredibly important aspect of microglia development and function.

We thank the Reviewer for their appreciation of our work and for their constructive criticisms. We have made modifications to the introduction and text, and added a new illustration to help make the story more accessible and coherent to a broader audience. We have also greatly expanded our inclusion of representative control images as well as quantification and statistical analyses in our paper. In addition, we included quantification and analysis of microglia phenotypes in males vs. females. We designed our breeding strategy based on recommendations from the Genetic Resource Science group at The Jackson Laboratory and published by Luo et al.⁷. All WT, floxed, and recombined alleles were assessed by genotyping. We note evidence of parental sex bias in germline recombination in *hGFAP^{Cre}* mice. With our breeding strategy, there were no true “wild type” mice (without transgenes or floxed alleles). Rather, comparisons were generally made between fl/+;Cre (control) and fl/fl;Cre (mutant/case). We also compared littermate controls from one litter to another and found no appreciable difference. Based on this, and to reduce redundant documentation, we have included a single representative control in each figure cohort. We have added this information into the Methods section.

Altogether, we feel that the current manuscript is much stronger thanks to the Reviewer’s comments, and that our main arguments have been bolstered by the newly included data.

Major comments:

*1. There is a lack of image quantification across the entire manuscript that makes the results difficult to interpret. For example, in Figure 1 there is no validation of the *Itgb8* knockout in the specific target cell populations, if this is consistent across the brain or even where the images were taken from. There are no wildtype littermate controls. Figures 4, 5 6 and 7 also require quantification for all stains. For example, in figure 4, what is the percentage of deletion in the different *Tgfb1* flox line manipulations? The vast majority of the imaging based figures are similar in their lack of proper controls, quantitative data and statistics.*

We greatly appreciate these suggestions, which are also in line with a major point by Reviewer #2. We added the following quantifications as well as new data and analyses in response to this important comment:

- Validation of *Itgb8* knockout and quantification of deletion efficiency: We performed western blot analyses of ITGB8 expression in *Emx1^{Cre};Itgb8^{fl/fl}* mutants compared to *Emx1^{Cre};Itgb8^{fl/+}* controls, as well as in *hGFAP^{Cre};Itgb8^{fl/fl}* mutants compared to *hGFAP^{Cre};Itgb8^{fl/+}* controls (n = 3 for each genotype). We quantified these western blots and performed statistical analyses of the data (see Supplementary Figure 4).
- Inclusion of controls: We included additional representative control images to the manuscript (see Figures S2, S4 and S9, and above explanation of controls).
- Quantification of staining: We included quantification and statistical analyses each genotype (at least n = 3) for the various immunohistochemical stainings we performed in Figures 1, 2, 4, 5, and 7. We also included quantification and statistical analyses for Supplementary Figures 4, 9, and 11.

For more details on various new quantification and analyses, please also see the description provided in response to Reviewer #2's major point 4.

*2. The figures often do not substantiate the claims made in the figure legend or text. For example, Figure 1 does not demonstrate that microglial maturation is dependent on interactions with radial glia. There are two example images, no quantification, no statistics and no genotype differences shown. How common are the interactions between embryonic radial glia and microglia precursors (Figure 1H)? Quantitative data demonstrating that these contacts happen more often than expected by chance would be required to make this claim " In total, these data suggest that *Itgb8* expression in early embryonic radial glia is necessary for the TGFb-dependent differentiation of CD206+ microglial precursors, and that in the absence of *Itgb8*, these CD206+ precursors populate the brain, but fail to express microglial-specific markers."*

This is similar to the response to Reviewer 2, Major Comment 3. Please see our response above, in addition to our answer here.

We would like to point out that while we agree with Reviewer 3's point that we have not proven that radial glia make preferential contact with CD206+ meningeal macrophages, in our view it is not necessary for the precursors to make preferential contact, but rather that they have *some* contact with radial glial endfeet. Radial glia endfeet make very dense projections to the meningeal layer where the majority of embryonic CD206+ macrophages reside⁸. Furthermore, we demonstrate the presence of end-feet contact with CD206+ cells using the *Emx1^{Cre};RG-Brainbow* model. We contend that even if occurring by chance (as opposed to preferentially) radial glia:CD206 interactions are sufficient to activate ITGB8-dependent TGFβ signaling and promote microglial differentiation.

It would also be necessary to actually look at additional timepoints to demonstrate that the CD206+ precursors do populate the brain as expected but do not show microglial markers.

We appreciate and agree with the reviewer's request for additional timepoints to examine the population of the brain with immature CD206+ macrophages. We previously detailed *Itgb8^{fl/fl};Nestin^{Cre}* and *Tgfb2^{fl/fl};Cx3cr1^{Cre}* mutant mice at E16, P7, P15 and P30 for various markers of microglial dysmaturation including APOE, P2RY12, TMEM119, and refer the reviewer to this published work¹. In the present work, we found similar phenotypes in *Itgb8;Emx1^{Cre}* and *Tgfb1* null mutants at E14.5 and in adulthood, which is consistent with our original interpretation that dysmature microglial cells populate the entire CNS similar to homeostatic microglia.

Is it possible that these precursors are actually BAMs?

As the referee suggests, our model is that dysmature microglia are BAM-like, in that they lack expression of markers that differentiate microglia from a BAM-like precursor expressing canonical BAM markers including CD206. As noted above in response to Reviewer 2, research from the Prinz and Miyata groups indicate that microglia are derived from a BAM-like CD206 progenitor^{5,6}. Interestingly, while dysmature microglia in *Itgb8* mutant models do show expression of perivascular-, meningeal- and choroid plexus-associated BAM genes, we have not found that dysmature microglia have preferential association with these regions.

As another example, in the discussion the claim is made that "Our analysis shows that this is reflected by widespread epigenetic differences in homeostatic, MGnD/DAM and BAM gene loci. These epigenetic differences are long-lasting and stable through adulthood, beyond the point at which Itgb8 appears to be necessary for microglial homeostasis." This isn't shown in the paper as there was only an adult timepoint done for analysis and it is not clear when the epigenetic differences arise. These types of unsubstantiated claims occur throughout the manuscript.

We appreciate the reviewer's interpretation. Indeed, our primary observation is that these epigenetic changes are present in adulthood. We have changed our language in the manuscript to reflect this.

3. The presentation of the transcriptomics and epigenomics datasets are not clear. For example, is the heatmap in Figure 2 scaled in some way? The legend lacks labels and it is unclear how the statistics were done to assess significant enrichment throughout this figure. Are these gene expression changes unique to developmentally regulated genes or are other genes also misregulated? It is unclear if this data was generated for this paper or retrieved from GEO from another publication. If this is the first analysis of the dataset, a standard differential gene expression analysis would be the appropriate first step in the analysis. There is no statistical testing of how the control and Itgb8 microglia change in the target gene expression in Figure 2H. Statistics need to be clearly defined that include corrections for multiple comparisons and significant p-values for individual genes reported.

We thank the reviewer for their feedback and apologize for the lack of clarity here. We have updated the figure, figure legend and main text to address each of these concerns. All the sequencing data used in Figure 2 were obtained from GEO accession GSE239603, and were initially described in along with *Smad2/3* sequencing data⁹. We now include this information explicitly in the text describing these results. As a clarification, all datasets were re-analyzed with our pipeline (Rsubread → FeatureCounts → DESeq2), regardless of if they were newly generated or from a previous work.

The heatmap in Figure 2A shows all genes previously described to be developmentally regulated in mouse microglia¹⁰ not only those differentially expressed. Note that we did not evaluate enrichment of any cluster, because we did not subset the dataset to only DE genes. We decided to construct it this way in an effort to show these results in a completely unbiased way, even at the expense of using relatively noisy genes.

The data was normalized in two steps using standard procedures, as follows: raw counts were normalized using DESeq2 (to account for library size and RNA composition), and then z-scores were calculated. The heatmap shows these z-scores of normalized expression. The revised version of the figure states “z-score” as the unit in the scale label. This information has been added to Supplementary Figures 6B and 8A, which we noticed lacked this label thanks to the reviewer’s suggestion. A similar procedure was performed for epigenomic datasets; we now have added “z-score” to the scale in Figure 3D.

Similar to above, bar plots in Figure 2B were from the data for all genes in the heatmap (not only those differentially expressed). They indicate mean and standard deviation of the normalized expression (by the DESeq2 procedure) of all genes in each cluster.

The revised version of Figure 2H now indicates genes that showed statistical significance (at $FDR < 0.05$) with an asterisk.

The heatmap as a whole was constructed with all the *bona fide* genes in each set, as described previously (DOI 10.1038/s41590-023-01627-6). Thanks to the reviewer’s comment, we were able to detect a mistake in the scale label. It now reads “z-score” instead of “Log2FC”.

Similarly, for Figure 3 it is unclear how widespread the chromatin accessibility and H3K9ac differences are in the wildtype compared to Itgb8fl/fl;Emx1Cre dysmature microglia. Having a standard volcano plot with significant regions depicted would be required to visualize the extent of the chromatin and acetylation changes.

We thank the reviewer for their suggestions regarding the presentation of the epigenetic analysis. We have generated volcano plots to display the ATAC-seq and H3K9ac results; these plots are located in Supplementary Figure 4D.

It is also difficult to access the magnitude of change in the heatmap in 3D and 3E as they appear scaled in some way that is not clearly defined in the legend. The correlations in Figure 3B are also hard to read as many of the gene names overlap.

We apologize for the lack of clarity regarding the heatmap in 3D and 3E. We clarified the scaling effect by labelling all heatmaps with “z-score”. Scaling by row, instead of total counts for example, for each gene allows for uniform representation of expression differences between conditions, which is standard in heatmap presentation. The overlapping gene names in 3B were also observed by Reviewer #2. We have updated Figure 3 to address these concerns.

Please also include all differential gene, region and acetylation lists in a supplemental excel file to make it easy for readers to examine genes and regions of interest. A UCSC browser track hub or bigwig files on GEO would also help for better data access for readers.

We have included in our resubmission a supplemental excel file containing all ATAC-seq and H3K9ac peaks and their associated gene and genomic region coordinates. As part of our initial Nature Communications submission, we previously created a UCSC browser track hub that can be easily used to visualize the ATAC-seq and H3K9ac data.

Similarly, for Figure 7 it is unclear how Figure 7C was constructed. Are the correlations done in terms of log fold change in expression differences between wildtype and mutant for each condition? Or some other metric? It would be nice to see how many differentially expressed genes are shared or unique among the mutants – potentially using something like an upset plot. Figure 7E lacks a legend and statistics. Do any of the genes

represented reach statistical threshold for significance after FDR correction? The overlaps with ATM and PAM as mentioned in the text are not shown.

We agree with the reviewers that we can provide more clarity about how Figures 7C and 7E were generated. We have added descriptions to our Figure Legend and Methods section to address these concerns and explain it below.

In Figure 7C, we constructed the correlation matrix using the Pearson's correlation between log2FC as estimated by the DESeq2 model (it builds a generalized linear model assuming the raw count data follows a negative binomial distribution). We decided to use this parameter because a) it includes the normalization procedure; and b) it summarizes the trend of all samples in a given group in relation to their own control. To define the set of genes contributing to the correlation, we used the union of the sets of genes differentially expressed in at least one mutant vs. its control (the final set consisted of 4406 genes).

We are thankful to the reviewer for their useful suggestion of using a upset plot to show shared DE genes in all mutants. These plots are now included in Supplemental Figure 8.

We apologize for our lack of a legend in Figure 7E. We have added a legend "z-score" and a description of the statistical analysis to the Figure Legend. The heatmap shown in Figure 7E was constructed using normalized expression of whole brain datasets, scaled to z-score. Since some samples do not share the same controls (i.e. *Lrrc33* mutants were from a previously published study; *Itgb8* mutants require *Nestin^{Cre}* controls while T1, TR2, and S2/3 require *Cx3cr1^{Cre}* controls), to compress the information into a single heatmap we averaged all the controls of each type of plot each in one column. Thus, there are 3 Control rows in the heatmap: *Nestin^{Cre}* controls, *Lrrc33* controls, and *Cx3cr1^{Cre}* controls - all statistical analyses use the original number of samples and is performed against the respective controls, so the data treatment in this heatmap is only for visualization purposes.

In the revised version of Figure 7E, we now include a legend and a visual indication of statistical significance to show DE genes in the heatmap. Since this heatmap shows large amounts of information in a compressed format, we decided to indicate DE genes in each mutant with a dashed rectangle framing these genes (we refer to genes as DE when statistically different from the respective control). To avoid extreme clutter in the graph, we only framed genes that were DE in all mutants (vs. their respective controls). Note that DE is tested with DESeq2 and includes FDR control. We hope that this way of showing significance is sufficiently clear without interfering with the understanding of the information within the heatmap.

Regarding the use of ATM and PAM genes in the text, we wanted to point to the reader that the genes of interest *ApoE*, *Lgals3*, and *Clec7a* -and those specifically- do not only mark DAM/MGnD microglia, but also developmentally restricted microglial populations with different physiological functions. For this reason, we mentioned ATM/PAMs, but we did not include a directed analysis of the expression of *bona fide* markers of these populations.

4. The writing is very field specific, and jargon filled making it difficult to read and unclear what the overall significance of the work is for readers in a broader audience. In the introduction, a first paragraph on the large importance of this area of work would be helpful for framing the study for the diverse readership of Nature communications. The introduction is also incredibly dense and very full of jargon. For example, it isn't clear that avB8 is the same as Itgb8.

We thank the reviewer for this comment and agree that additional introductory remarks as well as additional explanation/simplification of field-specific language will help the paper be accessible to a broader audience. We rewrote the beginning of our introduction to help frame the study for the Nature Communications audience. We also tried to clarify more “niche” or specialized language throughout, as with the $\alpha V\beta 8$ /ITGB8 example described above by the reviewer.

The figure diagram in figure 7A helps clarify this but doesn't come until the end of the paper. Moving the panel from 7A to figure 1 and adding in the canonical vs non-canonical signal details would help. There are also lots of different transgenic manipulations and staining conditions across these figures and better labeling and/or diagrams would help readers follow the logic of the conditions more easily.

We thank Reviewer 3 for again making these insightful suggestions. We generated a new graphical abstract to illustrate the following points: 1) As embryonic microglial precursors reach the brain, TGF β signaling is mediated by the interactions between neural progenitors and microglia, and is crucial for the proper development of microglia. In our study, we have characterized the common microglial phenotype present in many TGF β pathway mutant models (genetic disruptions shown by red “X”). 2) Developmentally arrested “dysmature” microglial persist into adulthood with unique transcriptional/epigenetic characteristics. 3) Mice with developmental deletion of downstream TGF β pathway genes in microglia have dysmature microglia whose expression of reactive markers correlate with disease severity.

We have also added a new summary illustration in Figure 6 to act as a guide for readers.

The graphical abstract also implies that the work is heavily focused on microglia-radial glia interactions in development which is misleading given this is only one panel of figure 1 and there isn't any quantitative data on this point.

In addition to our studies herein, it was conclusively shown via genetic fate mapping that microglia are derived from a CD206⁺ precursor population; the majority of CD206⁺ macrophages reside in the meninges during embryonic development³. We believe the combination of this previous work, along with our genetic studies and images of radial glia endfeet interactions with microglia precursors work together to build the model we have presented in the graphical abstract.

5. Sex is not included as a biological variable in any of the experiments. Sex has an important influence on microglial development (see PMID: 31743527 for a review) and is a critical mediator of disease risk. The sex of the animals is not stated for any of the experiments and all experiments should have both male and female animals included and analyzed for sex differences.

We thank the reviewer for this comment. As noted in our response to Reviewer #2's comment regarding study rigor, we performed all experiments on both sexes with approximately equal numbers of age-matched mice in adults, and approximately equal distribution in embryos. Despite the important role of sex in microglial development, we did not detect significant phenotypic differences among male vs female mice. In our whole brain RNA sequencing analysis, we did not observe a correlation between sex and the expression of MgND and homeostatic genes. Finally, we present new data of adult male and female *Lrrc33* knockout mice as a representative model to explore potential sex-specific effects in microglial phenotypes (new Supplementary Figure 10). By immunostaining for various microglial markers (IBA1, P2RY12, CD206, and LGALS3) in male vs. female *Lrrc33* knockout and control mice (n = 3 of each condition), we find that there is no evidence for major differences in phenotype based on sex.

Minor comments:

1. In multiple figures, the dotted blow-up box doesn't match the small box. Everything needs scale bars and pink boxes over red staining is not easy to see. There are also multiple problems with the labeling of figures and legends.

We apologize for the noted inconsistencies. We have adjusted boxes and insets, added scale bars, and adjusted the color scheme for various figures as suggested. We have also made sure our figure legends and figures use consistent labeling and are in alignment with each other.

2. Figure 2A. A white, aged mouse is shown in the figure, but black mice were used throughout the study. Suggest replacing for clarity.

We agree that a black mouse would be more appropriate here and have changed the mouse color to black.

3. The figure legend for Figure 2 appears to be wrong, the title doesn't match the text and the individual panels are not described sufficiently.

We thank the reviewer for catching this and apologize for the inconsistency. We have updated the figure legend title. We have also significantly expanded the descriptions in this figure.

4. In Figure 4A, what causes the vascular defects and hemorrhage? weaker expression in brain blood vessels (both endothelial cells and PDGFR β + mural cells) at E14.5.

We thank the reviewer for this intriguing question. We document a cell-autonomous effect for *Tgfb1* in microglial development, but we were not able to determine a sole cellular source of *Tgfb1* responsible for maintaining vascular integrity. We propose that *Tgfb1* expressed from endothelial cells can compensate for the loss of *Tgfb1* from mural cells, or vice versa.

5. In Figure 6, is the patchiness of Tgfb1 loss of expression due to failure in recombination? Please quantify the number of microglia with recombination and compare phenotypes of recombined and non-recombined microglia quantitatively for the different stains.

We thank the reviewer for this suggestion. Yes, we believe that the patchiness is due to lack of recombination, as the "intact" patches have retained *Tgfb1* expression by RNAscope. We refer the Reviewer to our recently published paper, which has quantification for the recombination efficiency for multiple microglial CreER lines¹¹.

6. Figure 7F requires quantification of LGALS3 levels, preferably by more quantitative methods such as western blot on sorted microglia or intracellular flow cytometry.

We thank the reviewer for their feedback regarding LGALS3 quantification. Per this request and the request in their Major Point #1, we have added a Supplementary Figure that contains a whole brain western blot quantification of LGALS3 expression changes in *Cx3cr1^{Cre};Tgfb1^{fl/+}* controls, *Cx3cr1^{Cre};Tgfb1^{fl/fl}* mutants and *Cx3cr1^{Cre};Smad2/3^{fl/fl}* mutants. Consistent with our histological analysis, we found a significant 3.4-fold increase in LGALS3 protein expression in *Cx3cr1^{Cre};Tgfb1^{fl/fl}* mutants as compared to *Cx3cr1^{Cre};Tgfb1^{fl/+}* controls. We saw no statistically significant increase in expression by western blot analysis in *Cx3cr1^{Cre};Smad2/3^{fl/fl}* mutants compared to controls, suggesting that the presence of LGALS3+ cells in the white matter tracts of *Cx3cr1^{Cre};Smad2/3^{fl/fl}* mutants is not sufficient to show statistically significant overall changes in whole brain LGALS3 expression by this quantification method.

7. Neuromotor phenotypes are missing from Figure S8 and the immune staining figures lack quantification.

We thank the reviewer for this observation. We did not observe any rescue of the motor phenotype in *Cx3cr1^{Cre};Tgfb1^{fl/fl};Apoe^{-/-}* mice, but we unfortunately did not make any quantitative measurements of this. In the interest of time, we have focused on adding quantification data for the major points mentioned by Reviewers #2 and #3.

8. The methods lack information on mouse husbandry including light cycle, diet etc. Control genotypes are not clearly stated for each experiment nor is information on breeding strategy.

We apologize for this oversight and have added details regarding husbandry conditions, control genotypes and breeding strategy. Please also see above description of controls.

9. The fluorescence-activated cell sorting is based on several microglial markers (two are explicitly listed in the methods, others are unclear). Are these markers significantly altered in the various mutants? Please include the sort data and yields to assess purity and potential impacts on microglial survival.

We apologize for not providing this data. Indeed, CD45 does change in its expression level, but our gating strategy collects macrophages at multiple levels of CD45 expression. We reported on our gating strategy, and the finding of increased CD45 expression, in our previous publications (^{1,9}) and refer the reviewer (and reader) to these publications for a deeper description.

References:

1. Arnold, T.D., Lizama, C.O., Cautivo, K.M., Santander, N., Lin, L., Qiu, H., Huang, E.J., Liu, C., Mukoyama, Y., Reichardt, L.F., et al. (2019). Impaired $\alpha V\beta 8$ and TGF β signaling lead to microglial dysmaturation and neuromotor dysfunction. *J. Exp. Med.* 216, 900–915. <https://doi.org/10.1084/jem.20181290>.
2. Arnold, T.D., Niaudet, C., Pang, M.-F., Siegenthaler, J., Gaengel, K., Jung, B., Ferrero, G.M., Mukoyama, Y., Fuxe, J., Akhurst, R., et al. (2014). Excessive vascular sprouting underlies cerebral hemorrhage in mice lacking $\alpha V\beta 8$ -TGF β signaling in the brain. *Development* 141, 4489–4499. <https://doi.org/10.1242/dev.107193>.
3. Arnold, T.D., Ferrero, G.M., Qiu, H., Phan, I.T., Akhurst, R.J., Huang, E.J., and Reichardt, L.F. (2012). Defective Retinal Vascular Endothelial Cell Development As a Consequence of Impaired Integrin $\alpha V\beta 8$ -Mediated Activation of Transforming Growth Factor- β . *J. Neurosci.* 32, 1197–1206. <https://doi.org/10.1523/JNEUROSCI.5648-11.2012>.
4. McKinsey, G.L., Lizama, C.O., Keown-Lang, A.E., Niu, A., Santander, N., Larphaveesarp, A., Chee, E., Gonzalez, F.F., and Arnold, T.D. (2020). A new genetic strategy for targeting microglia in development and disease. *eLife* 9, e54590. <https://doi.org/10.7554/eLife.54590>.
5. Masuda, T., Amann, L., Monaco, G., Sankowski, R., Staszewski, O., Krueger, M., Del Gaudio, F., He, L., Paterson, N., Nent, E., et al. (2022). Specification of CNS macrophage subsets occurs postnatally in defined niches. *Nature* 604, 740–748. <https://doi.org/10.1038/s41586-022-04596-2>.
6. Hattori, Y., Kato, D., Murayama, F., Koike, S., Asai, H., Yamasaki, A., Naito, Y., Kawaguchi, A., Konishi, H., Prinz, M., et al. (2023). CD206+ macrophages transventricularly infiltrate the early embryonic cerebral wall to differentiate into microglia. *Cell Rep.* 42, 112092. <https://doi.org/10.1016/j.celrep.2023.112092>.
7. Luo, L., Ambrozkiwicz, M.C., Benseler, F., Chen, C., Dumontier, E., Falkner, S., Furlanis, E., Gomez, A.M., Hoshina, N., Huang, W.-H., et al. (2020). Optimizing Nervous System-Specific Gene Targeting with Cre

Driver Lines: Prevalence of Germline Recombination and Influencing Factors. *Neuron* 106, 37-65.e5. <https://doi.org/10.1016/j.neuron.2020.01.008>.

8. Radakovits, R., Barros, C.S., Belvindrah, R., Patton, B., and Müller, U. (2009). Regulation of Radial Glial Survival by Signals from the Meninges. *J. Neurosci.* 29, 7694–7705. <https://doi.org/10.1523/JNEUROSCI.5537-08.2009>.
9. Yin, Z., Rosenzweig, N., Kleemann, K.L., Zhang, X., Brandão, W., Margeta, M.A., Schroeder, C., Sivanathan, K.N., Silveira, S., Gauthier, C., et al. (2023). APOE4 impairs the microglial response in Alzheimer's disease by inducing TGF β -mediated checkpoints. *Nat. Immunol.*, 1–15. <https://doi.org/10.1038/s41590-023-01627-6>.
10. Thion, M.S., Low, D., Silvin, A., Chen, J., Grisel, P., Schulte-Schrepping, J., Blecher, R., Ulas, T., Squarzoni, P., Hoeffel, G., et al. (2018). Microbiome Influences Prenatal and Adult Microglia in a Sex-Specific Manner. *Cell* 172, 500-516.e16. <https://doi.org/10.1016/j.cell.2017.11.042>.
11. Bedolla, A., McKinsey, G., Ware, K., Santander, N., Arnold, T., and Luo, Y. (2023). Finding the right tool: a comprehensive evaluation of microglial inducible cre mouse models. Preprint at bioRxiv, <https://doi.org/10.1101/2023.04.17.536878> <https://doi.org/10.1101/2023.04.17.536878>.

Please see below for our point-by-point response to the reviewers.

Reviewer #2: The authors addressed most of the reviewer's concerns and the modifications made including the new data have strengthened the manuscript and its conclusions. Nevertheless, some points need to be further addressed.

We are again grateful for the Reviewer's positive evaluation of our work and constructive criticisms. Please see below for our responses to these additional requests.

1. For Figure 1D-H, the authors should provide images of the control group that correspond to the quantitative data presented.

We have provided the images for control groups in Fig 1 (*Itgb8^{fl/+};Emx1^{Cre}* and *Itgb8^{fl/+};hGFAP^{Cre}*) in Supplemental Figure 4.

2. There appears to be an inconsistency between the quantitative data presented in Figure 2G and the corresponding figures 2E and F. It seems that the data for the cortex and striatum may have been reversed.

Thank you for noticing this mistake. We have corrected it.

3. There is a discrepancy in the numbering of the supplementary figures. Please verify and correct the numbering of the figure starting with "Gene Ontology and...".

Thank you for noticing this mistake. We have now numbered these figures correctly.

4. Please ensure that all references in the main text are updated accordingly.

All references are now updated.

Reviewer #3:

Summary: The manuscript is much improved from the previous submission but still requires several major and minor revisions before publication.

Thank you for your positive appraisal of our revisions. We have addressed the Reviewers additional remaining concerns below.

Major Concerns:

1. For lines 293-317, this experiment uses P30 mice, which are adolescent not adult. The manipulations were also done on P4-6 which is developmental, not adult. Claims about adult homeostasis would require manipulations in the adult mouse (P80 or older).

We appreciate the Reviewer's concern about adolescent versus adult in mice. For this manuscript, we will consider "adult" as mice P60-90 or older. For mice P30-60, we will consider these to be "adolescent" and have revised the manuscript accordingly.

2. Several figures still lack image quantification. For example, Figure 6 requires quantification.

We appreciate the Reviewer's request for quantification, and have added quantification throughout the manuscript where feasible and meaningful. In this case, we don't believe that additional quantification in Figure 6 would provide much value beyond what we've presented. As it stands, we observe stark differences in the properties of microglia that lack *Tgfb1* expression, as verified by RNAscope: they have a reactive morphology, expression of CD206, loss of the homeostatic markers P2RY12 and TMEM119, and loss of phosho-Smad3

staining. The images that we showed are internally controlled, as there are cells that are not recombined, and lack the changes seen in *Tgfb1* mutant microglia.

3. Figure S10 is missing the neuromotor data described in line 373 and the images require quantification to make the claim that there are [no] major impacts on microglial phenotypes (line 373).

We appreciate the Reviewer's additional requests for quantification, as it could provide a more nuanced picture of how in this case APOE might affect microglial and neuromotor functioning. We respectfully disagree with the Reviewer with regard to the stated "requirement" of quantitation in order to make our conclusions. We did not quantitatively assess neuromotor symptomatology, and unfortunately, do not have these mice in our facility any longer to make these assessments. Fortunately, the behaviors that were observed in these mice (hind limb gait dysfunction, intermittent seizures, piloerection, early death) are very easily apparent, and did not *obviously* change depending on whether or not the APOE gene is present. Similarly, while we did not quantitatively assess brain immunofluorescent images for potential subtle changes in microglial phenotypes, the lack of major and obvious changes in gross microglial morphology and marker expression (CD206 and P2RY12) associated with the presence or absence of APOE is easily apparent without quantification. We have adjusted the language to state the lack of quantitative assessment as a potential caveat to our observations, but would like to maintain our conclusions nonetheless.

4. Figure 7F-H requires quantification. The whole brain western (S11) is not sufficient quantification to draw conclusions about changes in the microglia.

We respectfully disagree with the Reviewer. *Lgals3* has low level expression at baseline, where it is largely found in border associated macrophages (Van Hove, et al. Nat Neurosci 22, 1021–1035 (2019)). We observed (and quantified) a significant increase in *Lgals3* mRNA expression in whole brains and microglia isolated from in *Itgb8*, *Tgfb1*, *Tgfb2*, and *Lrrc33* mouse models, and relatively reduced *Lgals3* expression in the *Smad2/3* mouse model. We quantitatively confirmed LGALS3 expression at the protein level using western blot (Supplementary Figure 11), which revealed no difference in LGALS3 expression in control vs *Smad2/3* mutants, compared to a significant increase in LGALS3 in the *Tgfb1* mouse model. We and others had previously observed that LGALS3 is preferentially associated with dense white matter tracks. We therefore performed immunofluorescence (IF) staining to non-quantitatively characterize the distribution of LGALS3 protein in the brains of the *Tgfb1* vs *Smad2/3* vs control mouse models, focusing on three different brain regions (cortex, and white matter tracts of the corpus collosum and striatum). Here, IF serves as an orthologous experimental approach to the more quantitative RNAseq and Western blot experiments, and confirms the dramatic and obvious increase in LGALS3 expression in microglia (IBA1 co-localized) in the *Tgfb1* mouse model, and roughly similar expression of LGALS3 in control and *Smad2/3* mouse models. In this case, we would argue that IF quantification is unnecessary when taken together with RNAseq and Western blot.

3. Sex is still not adequately addressed as a biological variable. Only three biological replicates are shown for each quantification, which is not sufficient to address sex differences. Please clarify which sex each data point is in each figure and discuss this as a limitation in the discussion.

To clarify, we performed all experiments on both sexes with roughly equal numbers of age-matched mice (n=3-10). However, we did not keep track of sex as a biological variable, as this was not a variable of interest to us, and unfortunately, it is not feasible at this time to retrospectively sex the samples as this sexing after the fact is done by PCR genotyping. As an alternative and based on the Reviewers previous requests, we took advantage of the *Lrrc33* mouse model which we currently have breeding in the lab, to determine whether sex affects the TGF β -dependent microglial phenotype. Using 3 mice of each sex and genotype, we find significant differences in mutants versus controls, and no apparent difference within male or female groups. We also assessed *Xist* gene expression (males) in whole brain RNAseq to retrospectively sex the mice used in RNAseq experiments and found no association between sex and transcriptional phenotype. We would respectfully disagree with the Reviewer's contention that 3 replicates are not sufficient to demonstrate the lack of sex-associated phenotypes that we observe within these groups. In fact, to rule out sex-specific effects within such diametrically different phenotypes only requires 2-3 mice (n=2.4) of each sex according to t-test power calculation using R with

conservative parameters: power=0.8, delta=0.8, standard deviation=0.2. Other published reports use similar N to diagnose lack of major sex-related changes in microglia (Thion et al. Cell, 2018). Furthermore, a recent publication by Bedolla et al (Nature Communications, 2024) showed no meaningful effect of sex on phenotypes due loss of TGFβ signaling in microglia, stating “we did not observe any sex differences in our iKO [TGFβ or ALK5] mice using immunohistochemical, transcriptomic, or behavioral analysis”, which is entirely consistent with our observations. We appreciate the perspective of the Reviewer who again pointed to the potential importance of sex as a biological variable in microglial development and homeostasis, but we see no evidence of this in our genetic models at this point.

Minor Concerns:

1. Please revise the wording in Supplemental figure 2. Brain-wide deletion of *Itgb8* with NestinCre results in loss microglial of microglial homeostasis and increased reactivity, as shown be loss of P2RY12 expression and upregulation of the reactive marker LGALS3.

This has been fixed. Thank you for noticing this mistake.

Supplemental figure 4L, the open arrow in the #2 blow up is not obviously pointing at an IBA1+ cell.

We have now placed the tip of the arrow closer to an IBA1+ cell.

To strengthen the claim that the dysmature microglia are similar to immature microglia, it would help to compare the ATAC-seq signatures to one of several developmental profiling studies with ATAC-seq across early microglial development such as (Matcovitch-Natan et al., 2016; Thion et al., 2018).

We thank the reviewers for this suggestion. We have added new data to Supplemental Figure 5E that compares ATAC-seq identified regions of open chromatin in the *Itgb8^{fl/fl};Emx1^{Cre}* cortical microglia versus early developmental stages, as described by Matcovitch-Natan et al., 2016. In support of our model, and in line with our RNA-seq analysis, we found a high degree of correlation between *Itgb8^{fl/fl};Emx1^{Cre}* cortical microglia and yolk sac microglial progenitors (R= .71, p<2.2e-16), a moderate degree of correlation with E12.5 early embryonic microglia (R= .47, p<2.2e-16) and a lower degree of correlation with “Pre microglia” (R= .28, p<2.2e-16), which Matcovitch-Natan et al., 2016 categorize as microglia from E14.5 to “a few weeks after birth”, although in their methods they say that their “Pre Microglia” dataset is derived from P3 microglia.

It is less clear that the epigenetic changes are “associated with MGnD and BAM gene upregulation” as stated in the title of this section (line 210-211). Please either revise the language and remove the adult part or provide evidence in adults.

We apologize for any confusion. We do not claim to have demonstrated that the observed epigenetic changes *cause* the associated changes in MGnD and BAM gene regulation – they may or may not. We intend only to draw the reader’s attention to our observations showing that there are epigenetic marks in gene regions which control expression of MGnD and BAM genes. There is in fact a clear *association* between epigenetic modification (ATAC and H3K9ac differentially accessible peaks (DAPs)) and genes that are differentially expressed in *Itgb8^{fl/fl};Emx1^{Cre}* versus control mice. Regarding the developmental timing of this association: we performed RNA-Seq and epigenetic studies on P60-P90 adult mice.

Figure 4C and D, please check the arrow placements on the blow-outs. They should be consistent across panels, but they are not.

These are fixed.

Line 288 references 5G. please double check that this is correct.

Thank you for noticing this mistake. This error has been fixed.

Line 297: “(how much intragastric)” should be an actual amount given.

Thank you for noticing this mistake. This error has been fixed.

Line 302. What are “b1-intact microglia”?

This was meant to be “*Tgfb1*-intact”. This error has been fixed. Thank you for noticing this mistake.

There are two supplemental 7 labelled figures.

This error has been fixed. Thank you for noticing this mistake.

Reviewer #4 (Remarks to the Author):
